# Quartz under stress: Raman calibration and applications to geobarometry of metamorphic inclusions

Bruno Reynard[1], Xin Zhong[2]

[1]Laboratoire de Géologie de Lyon, Univ Lyon, ENS de Lyon, Univ Lyon1, CNRS, France
[2]Institute of Geological Sciences, Freie Universität Berlin, 12249 Berlin, Germany

*Correspondence to*: Bruno Reynard (bruno.reynard@ens-lyon.fr)

**Abstract.** An experimental calibration of the shifts of three major Raman peaks of quartz with hydrostatic pressure and uniaxial differential stress is presented, and implications for their use in geobarometry based on Raman spectroscopy of quartz inclusions is discussed. The position of 206 cm$^{-1}$ peak depends only on hydrostatic pressure $P$, and its pressure

dependence is recalibrated with a peak fitting procedure that is more adequate for Raman barometry than previous calibrations. The position of the 128 and 464 cm$^{-1}$ peaks depends on $P$ and also on differential stress $\sigma$, which can be determined from the position of these two peaks knowing hydrostatic pressure from the position of the 206 cm$^{-1}$ peak. The results obtained here are different from those inferred previously from first-principles calculations. The present calibration provides direct relationships between Raman shifts and stress, with a simple formulation of residual pressure and differential

stress assuming uniaxial stress along the *c*-axis of quartz inclusions. It is tested on data from experimental and natural inclusions. Residual pressures from the present calibration are similar within uncertainties to those obtained with previous experimental calibration. Residual differential stresses obtained from the 128 and 464 cm$^{-1}$ peaks are very sensitive to the precision of Raman measurements. Experimental inclusions yield residual pressures consistent with synthesis pressure. Differential stresses obtained on some experimental inclusions are sometimes incompatible, providing a criterion for

identifying inclusions under complex stress conditions that are not appropriate for geobarometry. Recent data on natural inclusions show self-consistent differential stress, consistent with the assumption of major stress along the symmetry axis of the inclusion crystals. The average pressure values from the 128 and 464 cm$^{-1}$ peaks is similar to the residual pressure from the 206 cm$^{-1}$ peak that depends only on hydrostatic pressure. It can be used to obtain pressure when the 206 cm$^{-1}$ peak position cannot be used due to interference with host mineral peaks. Using the 128 and 464 cm$^{-1}$ peaks alone, or averaging

either 128 and 206 or 206 and 464 cm$^{-1}$ peaks can induce systematic bias in the residual pressure determination. Applications of the present results to natural inclusions suggest that combined determination of residual pressure and differential stress may be used both for barometry and thermometry pending further calibration.

# 1 Introduction

Raman spectroscopy has been used over two decades for determining residual pressures in mineral inclusions in diamonds (Izreali et al., 1999) and metamorphic minerals, principally garnets (Enami et al., 2007; Parkinson and Katayama, 1999). Entrapment pressures calculated from thermo-elastic modeling provide a determination of metamorphic pressures (Angel et al., 2017a; Kohn, 2014; Zhong et al., 2019; Zhang, 1998) that is independent of those obtained from metamorphic phase equilibria through thermodynamic modeling (Connolly, 1990; Holland and Powell, 1998). The method has been validated

with experimental entrapment of quartz in garnet at controlled hydrostatic pressures (Bonazzi et al., 2019; Thomas and Spear, 2018).

Quartz is often used because it is a common mineral of almost pure composition hence the position of its Raman peaks depends on intensive parameters and not on chemical variations. Accurate determinations of residual pressures rely among others on precise determinations of the Raman peak shifts with pressure that are provided by diamond anvil cell (DAC)

experiments (Schmidt and Ziemann, 2000). Differential stresses also influence the calculations of residual pressures, and determinations of strains from Raman frequency of quartz inclusions have relied on first-principles calculations of Raman peak shifts (Murri et al., 2019), a method that was also successfully used for determining differential stresses in high-pressure experiments (Reynard et al., 2019), but that is backed only by experimental data at liquid helium temperature (Briggs and Ramdas, 1977; Tekippe et al., 1973).

New experiments were performed to calibrate directly the Raman peak shifts induced by hydrostatic and non-hydrostatic stresses. Consequences for barometric applications are discussed based on available data on artificial inclusions from piston-cylinder experiments (Bonazzi et al., 2019), and on recent datasets on natural quartz inclusions in different suites of metamorphic rocks (Cisneros et al., 2021; Gonzalez et al., 2019; Zhong et al., 2019).

# 2 Experiments

Raman spectra under stress were obtained in the back-scattering geometrys using a Horiba™ HR Evolution (532.31 nm excitation from a DPSS laser) and 1800 gr/mm at LGL in ENS de Lyon. Only the three most intense Raman peaks at 128, 206 and 464 $cm^{-1}$ are analyzed, and will be referred to by these numbers in the reminder of the text. A series of experiments were first performed with calibration against silicon standard (crystals #1 to 4). In order to check for instrumental drift, sharp Hg 18312.55 or Ne 18516.59 $cm^{-1}$ lines (269.46 and 473.5 $cm^{-1}$ shift with respect to the excitation laser at 18786.05 $cm^{-1}$,

Fig. 1) were recorded along with all spectra in a second series of experiments (crystal#5 and higher). Maximum drift of the reference lines position during one experimental session is 0.5 $cm^{-1}$ and was systematically corrected. The measured separation between the two reference lines is in average of 204.03 $cm^{-1}$ with a maximum deviation of 0.15 $cm^{-1}$. The splitting of the two components of the 128 $cm^{-1}$ peak under stress perpendicular to the *c*-axis was determined by fitting two peaks with identical width. The incident polarization was rotated parallel and perpendicular to the c-axis of the crystal face

to enhance one component with respect to the other (Fig. 1). Quartz inclusions in eclogitic garnets from the Bergen Arc

(Zhong et al., 2019) were studied in Freie Universität Berlin. Raman spectra were obtained with a Horiba ISA Dilor Labram confocal Raman spectrometer using the 532 nm line of a Nd:YAG, 1800 gr/mm gratings, and Olympus 50× and 100× objectives. The laser energy was set at ca. 10 mW. Typical duration for each measurement was 1 to 2 min with >5 times repetition to obtain good signal to noise ratio. Immediately after each inclusion measurement, a gem-quality quartz crystal was measured as stress-free reference material to obtain the wavenumber shift of quartz inclusions due to residual stress.

Peak positions were fitted using symmetric Voigt profiles although some bands, in particular the 206 cm$^{-1}$ peak, are slightly asymmetrical in shape. This ensures the present calibration is comparable with data obtained on natural inclusions where Voigt profile must be used because interference with band of the host garnet do not allow to use more complex function to fit peak positions. Nominal frequencies of the three main quartz peaks of 127.1(2), 206.1(2) and 464.2(2) are obtained from internally calibrated spectra (Fig. 1). For the sake of comparison with previous studies (Bonazzi et al., 2019; Schmidt and Ziemann, 2000; Cisneros et al., 2021; Gonzalez et al., 2019; Zhong et al., 2019), all data were normalized to match ambient condition values of 128, 206, and 464 cm$^{-1}$.

Raman spectra of quartz were measured under hydrostatic pressure at ambient temperature on randomly oriented small crystals (<10 μm) in a DAC in a hydrostatic methanol-ethanol-water mixture (Reynard et al., 2015; Reynard et al., 2019) with small pressure steps up to 2 GPa. Pressure was measured with ruby fluorescence (Piermarini et al., 1975) for comparison with previous data (Schmidt and Ziemann, 2000). A Deben™ 5kN compression cell was used to compress at ambient temperature under uniaxial force parallelepipeds of quartz that were cut along or perpendicular to the *c*-axis, in a similar fashion to former experiments at 4 K (Tekippe et al., 1973). Crystals were placed between the two flat anvils of the system. A buffer made of an annealed 200-μm thick 301 stainless-steel foil was placed between the crystal faces and the anvils to lower the shear stress on the crystal ends through viscous flow of the metal. Apiezon grease was applied between anvils, metal foils, and crystals to reduce shear stress. The viscous grease helped maintaining the crystal stuck to one anvil while the system was being closed with a force <10 N (pressure <2 MPa) sufficient to hold it in place. Cutting orientation was performed optically with respect to crystalline faces of the initial crystal. Orientation of final rod faces with respect to simple crystallographic orientations were determined with conoscopic observations. The deviation from crystallographic axes was determined on unused rods with Raman intensities (Zhong et al., 2021b), and found to lie within 2° range (supplementary Fig. S1). The applied force is converted to stress by dividing with the cross section of the crystal rod, which was measured with an estimated uncertainty of 2%. A supplementary data sheet provides results from successful experiments (four out of twenty) and one example of unsuccessful experiment with early cracking of the crystal.

## 3. Elastic modeling

Interpretation of the Raman data on quartz inclusions relies on modeling of the residual pressure and differential stress for inclusions entrapped into a given host, generally garnet. The residual stresses are calculated based on an analytical model

(Zhong et al., 2021a) derived assuming a pure almandine garnet host. The model treats anisotropic inclusion entrapped in an infinite and isotropic host using the classical Eshelby's solution and the equivalent eigenstrain method (Eshelby, 1957; Mura, 1987). The unit cell parameters of quartz are fitted based on the X-ray data measured at ambient pressure high temperature (Carpenter et al., 1998), and at ambient temperature high pressure (Angel et al., 1997) with an EoS taking into account a curved alpha-beta transition (Angel et al., 2017b). The fitted results represent the axial EoS applicable to high P-T conditions considering the alpha-beta transition. For the PVT relationship of almandine host, we used the EoS from Milani et al. (2015). Entrapment stress is assumed to be hydrostatic. Due to the thermo-elastic anisotropy of the inclusion, the residual stress becomes non-hydrostatic after cooling and decompression. Here, compressive stress is defined as negative. A similar approach has been used (Alvaro et al., 2019) in dealing with different axial stress (strain) for quartz inclusion entrapped into an isotropic host.

Results show that expected residual pressures are generally positive for most crustal metamorphic conditions, and can be negative if the entrapment conditions are low pressures close and beyond the α–β quartz transition (Fig. 2). The residual differential stress is uniaxial with the symmetry axis of the stress parallel to the c-axis of the quartz inclusion. Stress on the inclusion is hydrostatic ($\sigma_c - \sigma_a = 0$) for a thermal gradient of ~220 °C/GPa or ~6 °C/km. Stress along the $c$-axis is higher than along the $a$-axis for higher temperature and lower pressure conditions, and lower for lower temperatures and higher pressures that are uncommon in metamorphic rocks. Residual pressure is more sensitive to entrapment pressure and residual stress to entrapment temperature (Fig. 2). A combined determination of the two has potential geothermobarometric application (Alvaro et al., 2019).

## 4. Results

### 4.1. Hydrostatic compression

The hydrostatic pressure effects were calibrated in the range 0-2 GPa that covers applications of Raman piezo-spectroscopy to metamorphic quartz inclusions (Fig. 3). For the 128 and 464 cm$^{-1}$ peaks, the present results are consistent with previous calibrations (Schmidt and Ziemann, 2000) with a maximum deviation of 0.4 cm$^{-1}$, and an average deviation of 0.1 cm$^{-1}$, i.e. within the inferred accuracy of Raman spectroscopy when internal standards provided here by Ne or Hg emission lines are used.

For the sake of comparison with former studies, data were fitted with the expression:

$$P = A\Delta\nu_h + B\Delta\nu_h^2 \tag{1},$$

where $\Delta\nu_h$ is the shift under hydrostatic stress with respect to the ambient conditions wavenumber ($\nu$) of the Raman peak. Values of A and B reported in Table 1 for the three main peaks of quartz. The Raman shift and its derivative are:

$$\Delta\nu_h = [-A + (A^2 + 4BP)^{1/2}]/2B \tag{2},$$

$(\partial \nu / \partial P)_o = (A^2 + 4BP)^{-1/2}$ (3).

It is worth noting that if the calibration is performed including data at pressures above 2 GPa, the initial slope is underestimated with respect to the present value, especially for the 206 cm$^{-1}$ peak. This is related to the complex pressure
dependence of quartz Raman spectrum beyond the 2 GPa pressure range (Morana et al., 2020; Hemley, 1987). Using our own data above 2 GPa (Reynard et al., 2019), we noticed a similar systematic deviation. The present calibration with its small pressure steps (15 points within 2 GPa) is particularly adapted for applications to quartz inclusions that are under residual pressures below 2 GPa, a condition that applies even to those that underwent maximum pressure conditions near the stability field of coesite with maximum residual pressure around 1.3 GPa (Alvaro et al., 2019; Korsakov et al., 2009).

**4.2. Uniaxial compression**

The effects of uniaxial stress, applied either parallel or perpendicular-to the *c*-axis of quartz, differ strongly from peak to peak (Table 1). Stress effects are linear within uncertainties within the investigated range of stress up to ~0.6 GPa beyond which crystals break. No symmetry breaking could be detected for compression along c-axis in spite of 2° deviation from the exact crystallographic direction (supplementary Fig. S1), hence analysis in the trigonal symmetry is valid. Symmetry
breaking was observed for compression perpendicular to the *c*-axis, marked by the splitting of the two components of the 128 cm$^{-1}$ E mode, in line with former experiments (Briggs and Ramdas, 1977; Tekippe et al., 1973).

For the 206 cm$^{-1}$ peak, the effects of stress are similar within uncertainties perpendicular and parallel to *c*-axis (Fig. 3c), with a stress dependence of 10.0(4) and 10.8(2) cm$^{-1}$/GPa, respectively. Because the shift of the 206 cm$^{-1}$ peak is independent of the stress orientation within uncertainty, its position gives an absolute reading of the pressure *P* whether the stress is
hydrostatic or not. Similar stress-induced shifts are obtained for crystal rods with different aspect ratio (1 and 4 along the *c*-axis, and ~2.5 and 4 along the *a*-axis), ruling out potential geometrical effects (supplementary Table). For the 128 and 464 cm$^{-1}$, the shift is very different when stress is applied perpendicular or parallel to the *c* axis. For stress applied along directions perpendicular to *c*, the two components of the 128 cm$^{-1}$ mode of E symmetry split due to the symmetry breaking (Tekippe et al., 1973). The splitting is maximum for uniaxial stress in the basal plane of quartz, a property that was used to
measure differential stresses in non-hydrostatic experiments with DAC (Reynard et al., 2019). The shifts of the 128 and 464 cm$^{-1}$ peaks relative to the 206 cm$^{-1}$ peak are linear within the investigated stress range (Fig. 4).

As a check of the internal consistency of the measured shift, the sum of axial stress-induced shifts (i.e. twice the shift perpendicular to the *c*-axis plus shift parallel to it) should compare with the slope of the bulk hydrostatic pressure dependence at zero pressure (Briggs and Ramdas, 1977). The sum of the axial stress dependences of the 206 cm$^{-1}$ peak is
30.8(10) cm$^{-1}$/GPa, similar to the bulk pressure induced shift of 30.7(13) cm$^{-1}$/GPa at ambient pressure, as well as for the 128 and 464 cm$^{-1}$ peaks, with values of 7.1 and 10.5 for the sum of axial stress dependences, and 7.9 and 9.4 cm$^{-1}$/GPa for the hydrostatic pressure dependence, respectively (Fig. 3, Table 1). The consistency between the sum of axial stress

dependence and hydrostatic pressure experiment suggests that the effect of symmetry breaking in uniaxial compression on stress-induced shifts is of second order.

## 5. Comparison with former studies

### 5.1. Hydrostatic compression

The present variation of the 206 cm$^{-1}$ peak shows an increasing discrepancy with that of Schmidt and Ziemann (2000) with increasing pressure, reaching about 1.5 cm$^{-1}$ at 1.5 GPa (Fig. 3a), above the accuracy of the measurement. This discrepancy is due to the use of a symmetric Voigt function in the present study when Schmidt and Ziemann (2000) used an asymmetric Pearson IV distribution. For piezospectroscopic applications, the Voigt function is widely used to fit the 206 cm$^{-1}$ peak position (Alvaro et al., 2019; Bonazzi et al., 2019; Cisneros et al., 2020; Cisneros et al., 2021; Enami et al., 2007; Gonzalez et al., 2019; Thomas and Spear, 2018; Zhong et al., 2019) because it avoid unrealistic fits due to interference with host mineral Raman peaks (garnet in most cases). Thus the present hydrostatic calibration of the 206 cm$^{-1}$ peak position is appropriate for common practice in quartz-inclusion Raman geobarometry. The pressure difference between the two calibrations is about 2%, it does not affect significantly the pressure estimates but improves the self-consistency in the calculations of residual differential stress on inclusions that is very sensitive to minute uncertainties (see below). Grüneisen plot (Fig. 3c) shows that the assumption of constant Grüneisen parameters (Angel et al., 2019) is not valid in quartz, except for the 464 cm$^{-1}$ peak. Grüneisen parameter ($\gamma$) dependence on volume is taken into account in fitting the data (Table 1) with two parameters (Reynard et al., 2012), the ambient pressure value $\gamma_0$, and its volume dependence $q$ expressed as $q=(\partial \ln\gamma / \partial \ln V)$. Pressure was converted to volume using equation of state (Angel et al., 1997). Non-linear Grüneisen parameters contribute to discrepancies between the present results and first-principles calculations on which constant $\gamma$ were fitted (Murri et al., 2019).

### 5.2. Non-hydrostatic compression

The relative shifts of the 128 and 464 cm$^{-1}$ peaks with respect to the 206 cm$^{-1}$ peak are remarkably constant between present experiments at 295 K and those at 4 K (Tekippe et al., 1973). Absolute values of stress-induced shifts are higher (~1.5 factor) in the present ambient temperature experiment than at 4 K where quartz is stiffer and has a lower volume corresponding to that of compression at about 0.25 GPa at ambient temperature. Shifts from the first-principles calculations (Murri et al., 2019) are compared to the experimental ones through the Grüneisen parameters listed in Table 1. To derive the Grüneisen tensor components from the present measurements, one needs to convert between stress and strain via the stiffness tensor. The Raman shift is related to stress as follows assuming $\sigma_1 = \sigma_2$ due to symmetry:

$$\Delta\nu = 2 \left(\frac{\partial\nu}{\partial\sigma_1}\right)_{\sigma_3} \sigma_1 + \left(\frac{\partial\nu}{\partial\sigma_3}\right)_{\sigma_1} \sigma_3 \tag{4},$$

where $\left(\frac{\partial v}{\partial \sigma_1}\right)_{\sigma_3}$ and $\left(\frac{\partial v}{\partial \sigma_3}\right)_{\sigma_1}$ are directly fitted from experiment (Table 1) and assumed constant. From linear elasticity, we have:

$$\sigma_1 = (C_{11} + C_{12})\varepsilon_1 + C_{13}\varepsilon_3 \tag{5},$$

$$\sigma_3 = 2\,C_{13}\varepsilon_1 + C_{33}\varepsilon_3 \tag{6},$$

where $C_{ij}$ are the components of the elastic stiffness tensor. The shear components of strain and stress are not present because for symmetry preserved crystal (i.e. $\sigma_1 = \sigma_2$), and the off-diagonal strain components are all zero due to $C_{41} = -C_{42}$. Substituting $\sigma_1$ and $\sigma_3$ into equation (4), and collecting common terms for $\varepsilon_1$ and $\varepsilon_3$, we have:

$$\Delta v = 2\left[\left(\frac{\partial v}{\partial \sigma_1}\right)_{\sigma_3}(C_{11} + C_{12}) + \left(\frac{\partial v}{\partial \sigma_3}\right)_{\sigma_1}C_{13}\right]\varepsilon_1 + \left[2\left(\frac{\partial v}{\partial \sigma_1}\right)_{\sigma_3}C_{13} + \left(\frac{\partial v}{\partial \sigma_3}\right)_{\sigma_1}C_{33}\right]\varepsilon_3 \tag{7}.$$

The Grüneisen tensor components are obtained by dividing the terms in square bracket by the peak position at zero stress $v_0$:

$$\gamma_1 = \left[\left(\frac{\partial v}{\partial \sigma_1}\right)_{\sigma_3}(C_{11} + C_{12}) + \left(\frac{\partial v}{\partial \sigma_3}\right)_{\sigma_1}C_{13}\right]/v_0 \tag{8},$$

$$\gamma_3 = \left[2\left(\frac{\partial v}{\partial \sigma_1}\right)_{\sigma_3}C_{13} + \left(\frac{\partial v}{\partial \sigma_3}\right)_{\sigma_1}C_{33}\right]/v_0 \tag{9}.$$

Using the values for $\left(\frac{\partial v}{\partial \sigma_1}\right)_{\sigma_3}$, $\left(\frac{\partial v}{\partial \sigma_3}\right)_{\sigma_1}$ obtained here and published values of $C_{ij}$ (Heyliger et al., 2003), we obtain the values reported in Table 1.

Shifts with uniaxial stress from first-principles calculations (Murri et al., 2019) are roughly consistent with experimental values for the 464 cm$^{-1}$ peaks, and divergent for the 128 and 206 cm$^{-1}$ mode. These differences are possibly due to shortcomings of the first-principles calculations such as unaccounted for anharmonic effects, or to assumption of constant Grüneisen parameters used in fitting theoretical results. Constant Grüneisen parameters are inconsistent with the pressure shifts of the 128 and 206 cm$^{-1}$ peaks (Fig 3c). It shows that the discrepancy between theory and experiments, although reasonable given the absence of fitted parameters in the first-principles models, is significant when trying to apply results to quartz piezobarometry. Thus we reexamine in the following the relationships between Raman peak positions and pressure and differential stress for quartz inclusions with the experimental shifts determined here.

## 6. Applications to piezospectroscopy of quartz inclusions

A spherical monocrystalline inclusion is subjected to triaxial normal stresses with principal components $\sigma_1, \sigma_2, \sigma_3$. The Raman shift $v$ is a function of these principal stresses and their orientation with respect to that of the crystal. Here we set $\sigma_3$ parallel to the c-axis of the inclusion. For the 206 cm$^{-1}$ peak, the shifts with differential stress are similar along and

perpendicular to the *c*-axis, and the Raman frequency will depend only on the residual pressure *P* defined as the first invariant of the stress tensor ($P = \frac{\sigma_1 + \sigma_2 + \sigma_3}{3}$). It is only for the 464 and 128 cm$^{-1}$ peaks that the different stresses will influence the Raman shift in response to both *P* and the differential stresses. Since the splitting of the 128 cm$^{-1}$ peak is not

observed in natural quartz inclusions, it is assumed that there is no significant difference between $\sigma_1$ and $\sigma_2$.

With a uniaxial differential stress along the major symmetry axis of the quartz inclusion, the stress tensor can be written:

$$\begin{pmatrix} P - 1/3\sigma & 0 & 0 \\ 0 & P - 1/3\sigma & 0 \\ 0 & 0 & P + 2/3\sigma \end{pmatrix} \qquad (10),$$

where $\sigma = \sigma_3 - \sigma_1 = \sigma_c - \sigma_a$ is the residual differential stress in the inclusion where *a* and *c* refer to the crystallographic axes of quartz. The shift $\Delta\nu_{nh}$ of a Raman peak of wavenumber $\nu$ with the differential stress is:

$$\Delta\nu_{nh} = \Delta\nu - \Delta\nu_h = 2/3 \ \sigma \ \left(\frac{\partial\nu}{\partial\sigma_3}\right)_{\sigma_1} - 2/3 \ \sigma \ \left(\frac{\partial\nu}{\partial\sigma_1}\right)_{\sigma_3} \qquad (11).$$

where $\Delta\nu_h$ is the hydrostatic shift from equation (2), $\Delta\nu$ is the measured shift with respect to ambient conditions, and $\left(\frac{\partial\nu}{\partial\sigma_i}\right)_{\sigma_j}$ are fitted slopes to the data under uniaxial compression. It is worth noting that while the shifts with pressure required fitting with a quadratic expression, the shifts with stress $\left(\frac{\partial\nu}{\partial\sigma_i}\right)_{\sigma_j}$ are assumed linear (Fig. 3 and 4, Table 1).

Combining (11) and (2) and solving for the stress, we obtain:

$$\sigma \approx 3/2 \ [\Delta\nu - (-A + (A^2 + 4B \times P_{206})^{1/2})/2B)]/[\ \left(\frac{\partial\nu}{\partial\sigma_3}\right)_{\sigma_1} - \left(\frac{\partial\nu}{\partial\sigma_1}\right)_{\sigma_3} \ ] \qquad (12),$$

where $P_{206}$ is the pressure obtained from the position of the 206 cm$^{-1}$ peak, and A and B are the fitted coefficients from the hydrostatic experiment given in Table 1. The non-hydrostatic stress can be obtained from the positions of both the 464 and 128 peaks, and checked for self-consistency. Using the values in Table 1, where for the 128 peak the average slopes for the two split components are used, stresses (in GPa) are obtained:

$$\sigma(464) \approx 1.5 \times [\Delta\nu(464) - (-0.107 + (0.107^2 + 0.0032 \times P_{206})^{1/2})/0.0016)]/(-2.7) \qquad (13),$$

$$\sigma(128) \approx 1.5 \times [\Delta\nu(128) - (-0.127 + (0.127^2 + 0.0204 \times P_{206})^{1/2})/0.0102)]/(1.8) \qquad (14).$$

To summarize, discrepancies between pressures obtained from the three Raman peaks are expected with increasing residual differential stress. The position of the 206 cm$^{-1}$ peak is independent of differential stress $\sigma$, and gives the hydrostatic pressure. If the residual differential stress is non-zero, residual pressures obtained from the 128 and 464 cm$^{-1}$ peaks should

be shifted with respect to values obtained with the 206 cm$^{-1}$ peak, by amounts of opposite signs. A routine for calculating pressure and stresses from the Raman peak positions is provided in the supplementary Excel spreadsheet. It is worth noting that the calibration established here translates directly the measured shifts of Raman peaks into stress, the geologically

relevant variable, while approaches using the Grüneisen tensor translate peak shift into strains that have to be converted to stresses using the elastic tensor.

As Raman measurements are subject to statistical fluctuations, so will the pressure (equation 1), and the two stresses calculated from equations (13,14) that therefore are likely to differ for a given inclusion (supplementary Table). Uncertainty in pressure will arise from uncertainty in the position of the 206 cm$^{-1}$ peak. Uncertainty on stress will arise from uncertainty in the position of the 128 cm$^{-1}$ and 464 cm$^{-1}$ peaks, and also in the position of the 206 cm$^{-1}$ peak that is used to calculate pressure in equations (13,14). A sensitivity analysis was performed to quantify the effect of small fluctuation in Raman peak

position on the calculated differential stress (supplementary Fig. S2). It is demonstrated that fluctuations as small as 0.2 to 0.4 cm$^{-1}$ in Raman band position can lead to a scattering of up to 0.5 GPa in differential stress. This explains the wide scattering of natural and experimental data as shown in Fig. 5 and 6. It also illustrates the origin of the anti-correlation between $\sigma(128)$ and $\sigma(464)$ observed in Fig. 5 and 6 from the larger uncertainty on the 206 cm$^{-1}$ peak position than on other peaks.

If a sufficient number of inclusions is measured, and if they belong to the same population (i.e. if they formed in a single event under similar pressure and temperature conditions), the average pressure and stress values can be obtained with a greater accuracy since the standard error of the mean will be greatly reduced in spite of the fluctuations. An alternative approach would be to use the over-determination of the system (2 unknowns and 3 independent equations or more if more peaks are used) to obtain the best-fit values of pressure and differential stress, or the strains (Bonazzi et al 2019, Murri et al

2019). It is our choice to compare the stresses obtained from equations (13,14) because their sensitivity to uncertainties allows discussing the self-consistency of the Raman measurements that may otherwise be blurred if a single best-fit value is calculated for each inclusion.

### 6.1. Comparison with experimental inclusions

  Bonazzi et al. (2019) synthesized quartz inclusions in almandine at 2.5 GPa 800°C and 3 GPa 775°C. The present

calibration was tested on their selection of inclusions based on optical criteria (absence of cracks, sufficient distance from garnet rim and other inclusions, ...). For most of these inclusions the residual pressures inferred from the 206 cm$^{-1}$ peak position are consistent with those expected from elastic modeling of quenching and decompression from the experimental equilibration conditions (Fig. 5a).

Several inclusions synthesized at 3 GPa display residual pressures lower than expected and that are inconsistent between the

three peaks (Fig. 5b), indicating large differential residual differential stresses (Fig. 5c). For those, the inferred residual differential stresses are of opposite signs when estimated using the 128 or 464 cm$^{-1}$ peaks, with values of $\sigma$ between ~ −0.5 and 2 GPa using the 464 cm$^{-1}$ peak, and between ~ 0.5 and −3 GPa using the 128 cm$^{-1}$ peak. It indicates these inclusions are under a stress pattern that is not consistent with simple elastic deformation. The large residual differential stresses are likely

due to unobserved defects around inclusions. Bonazzi et al. (2019) proposed a correction for similar effects in strains using the first-principles strain-induced shifts (Murri et al., 2019).

We propose to use the inconsistency in residual differential stress calculated here directly from Raman peak positions and equations (1,13,14) as a guide to eliminate those inclusions under complex stress state from the analysis of residual pressures. Thus we discarded from the analysis inclusions with an absolute difference in residual differential stress from the 128 and 464 cm$^{-1}$ peak positions larger than a value of 1 GPa defined from the sensitivity analysis (see supplementary Table and Fig. S2). This changes the average residual pressure from 1.06(3) to 1.11(3) GPa. Both residual pressure averages are within uncertainties of the value of 1.08 GPa expected from the elastic model. The average value of residual differential stresses after selection of 0.00(10) and −0.03(11) is close to that of 0.0 GPa expected from elastic modeling (Fig. 2), when the one prior to selection is ~ −0.5(2) and 0.4(2) GPa for the 128 and 464 cm$^{-1}$ peaks, respectively.

The inclusions synthesized at 2.5 GPa yield residual pressures of 0.82(1) after selection with the above-defined criterion from differential stresses using the 128 and 464 cm$^{-1}$ peak positions (Fig. 5b, supplementary table). They are consistent with those of the elastic model of 0.86 GPa. Residual pressures from the 128 cm$^{-1}$ peak are systematically lower by ~0.1 GPa. As a result, residual differential stress is −0.01(4) for the 464 cm$^{-1}$ peak, and of ~ −0.4(1) for the 128 cm$^{-1}$ peak, where a value of −0.05 GPa is expected from the elastic model at 2.5 GPa and 800°C (Fig. 2). The systematic shift of residual stress values from the 1:1 line (Fig. 5c) would correspond to a systematic deviation of only 0.5 cm$^{-1}$ in the 206 peak position for this particular set of inclusions, due for instance to interference with garnet peaks. The method use here allows detecting and illustrating the effects of such a minute deviation.

## 6.2. Comparison with natural inclusions

We used recent datasets on natural inclusions from three different metamorphic suites: 1) blueschists from Syros in garnet formed at ~1.4–1.7 GPa and 500–550 °C, and in retrograde epidote grown between ~1.3–1.5 GPa 400–500 °C and ~1.0 GPa and 400 °C (Cisneros et al., 2021); 2) Holsnøy eclogite from the Bergen Arcs (Norway) where inclusions formed at 1.4–1.6 GPa, 680–760 °C consistent with previous estimates based on phase equilibria (Zhong et al., 2019; Bhowany et al., 2018); 3) gneiss from Papua New Guinea with formation conditions estimated from Ti in quartz and quartz in garnet inclusions as ~1.0 GPa and 600 °C, which are interpreted as the P–T conditions of garnet growth and entrapment of quartz inclusions (Gonzalez et al., 2019).

The residual pressures estimated from the present calibration are within uncertainties of those published using earlier calibrations. The major difference comes from the possibility of estimating here the residual differential stress and to use it as a criterion for validating measurements (section 6.1). Residual pressures from the 128 and 464 cm$^{-1}$ peaks are respectively higher and lower than hydrostatic pressure from the 206 cm$^{-1}$ peak in rocks from Papua-New Guinea, indicating significant residual differential stress. They are almost indistinguishable within uncertainties for blueschists of Syros, indicating little residual differential stress (Fig. 6a, supplementary table).

Residual differential stresses $\sigma_{128}$ and $\sigma_{464}$ are self-consistent and low, with average values of less than 0.3 GPa (Fig. 6b, supplementary table). At these deviatoric stress levels, symmetry breaking effects are likely negligible (Murri et al., 2022). Average values are well defined except in inclusions from Norway where the 128 cm$^{-1}$ peak position is imprecise due to spectrometer configuration, and because of the small number of measurable inclusions. The dispersion of individual values is accounted for by uncertainties in Raman peak position (Figs. 6b and S2). The smaller dispersion of residual differential stresses in natural inclusions than in experimental ones likely reflects longer growth rates and potential anelastic relaxation times than in the relatively short experiments followed by fast quenching and pressure release. Such processes may affect the reconstructed initial P-T conditions and would deserve further investigation. With the method and calibration proposed here, all but five measured natural inclusions (see supplementary table) fall within the conditions defined for the calibration of uniaxial differential stress along the major symmetry axis of the quartz inclusion at the beginning of section 6. The method using strain anisotropy resulted in the rejection of 74 inclusions out of 92 (Gonzalez et al., 2019), a high rejection rate may stem from the different rejection criteria.

Average values of residual differential stress from both the 128 and 464 cm$^{-1}$ peaks are −0.04(4) GPa, −0.18(3) GPa, and −0.16(14) GPa for Syros, Papua-New Guinea, and Norway, respectively, with a similar trend to values expected from the elastic model, of −0.06(3), −0.13(3) and −0.14(4), respectively (Fig. 2). Taken at face values, residual differential stresses in Syros and Papua-New Guinea translate into temperatures of 340-600 and 640-830°C, respectively, to be compared with 500-550 and 600°C estimated from independent phase equilibria and geothermometers (Cisneros et al., 2021; Gonzalez et al., 2019). For the Holsnøy eclogite (Norway), the mean value of residual stress yields ~750°C within the range of 680-760°C from previous estimates (Zhong et al., 2019), although the temperature range is not precisely defined owing to the large uncertainty on residual stress. It suggests that independent constrain on temperature of entrapment may be obtained from residual differential stresses on inclusions. This would require reducing uncertainties by measuring residual pressures and differential stresses on a large number of inclusions, by systematic calibration of Raman measurements with Ne or Hg emission lines (section 2), and calibration on samples formed in well-characterized metamorphic conditions. The effect of large numbers of analyses on improving the accuracy of residual stress measurements is clearly seen when comparing mean values and uncertainties (Fig. 6b) from small (Norway, 11 inclusions), medium (Syros, 22) and large (Papua-New Guinea, 92) datasets.

## 7. Concluding remarks

The present calibration can be used to directly estimate the residual pressures and differential stresses using simple relationships established between Raman shifts, pressure and uniaxial stresses. A sensitivity analysis on calculated uniaxial stresses is proposed that allows detecting and evaluating systematic deviations in the Raman data. Accurate average pressure and deviatoric stress are obtained after checking that populations of inclusions belong to a single population formed at

similar conditions, and averaging over this population. It can be applied provided Raman peak positions are published in studies using elastic geobarometry.

Hydrostatic pressure is obtained from the 206 cm$^{-1}$ peak position that is insensitive to residual stress. Self-consistency of
residual differential stresses obtained from the 128 cm$^{-1}$ and 464 cm$^{-1}$ peak positions is an objective filter to detect inclusions that are under stress conditions where assumptions used for elastic modeling of entrapment conditions do not apply. Residual stress effects on the pressures determined with 128 cm$^{-1}$ and 464 cm$^{-1}$ peaks are of similar magnitude and opposite sign, hence the average of these two values is a good estimate of the hydrostatic pressure if the 206 cm$^{-1}$ peak cannot be measured accurately due to interference with host mineral peaks. Using the 128 and 464 cm$^{-1}$ peak alone, or
averaging either 128 and 206 or 206 and 464 cm$^{-1}$ peaks can induce systematic bias in the residual pressure determination. Residual differential stresses observed in suites of metamorphic rock inclusions suggest they depend on temperature at the time of entrapment. Reduction of uncertainties and calibration on rocks with various conditions of formation would be necessary for use as a quantitative geothermometer. Large numbers of inclusions should be measured and uncertainties on Raman data reduced with the systematic use of internal calibration lines. Reduction of uncertainties may also require Raman
measurements at combined high pressure and high stress, currently an experimental challenge, and elastic constant determinations at combined high pressure and temperature to limit extrapolation in elastic model predictions.

**Competing interests.** The authors have no competing financial interests or personal relationships that could have influenced the work shown in this publication.


**Acknowledgments.** This work was supported by grants to BR through LABEX Lyon Institute of Origins (ANR-10-LABX-0066) of the Université de Lyon within the Plan France 2030 of the French government operated by the National Research Agency (ANR). XZ is financially supported by the Alexander von Humboldt fellowship.

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

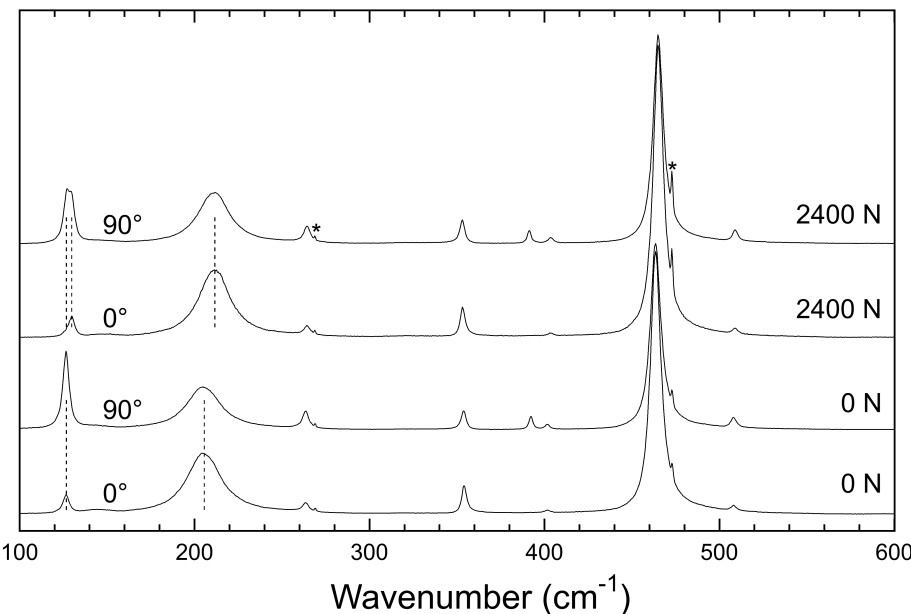


**Figure 1.** Typical spectra of quartz obtained in the back-scattering geometry at ambient conditions (0 N) and under uniaxial force of 2400 N (corresponding to a stress of ~0.6 GPa) perpendicular to the *c*-axis that appears vertical under the microscope. Spectra were measured with incident polarization parallel (0°) and perpendicular (90°) to the *c*-axis to better

characterize the splitting of the 128 cm$^{-1}$ peak under stress. No polarizer was put along the scattered light path. Asterisks mark the calibration Ne and Hg lines near 269.5 and 473.5 cm$^{-1}$.

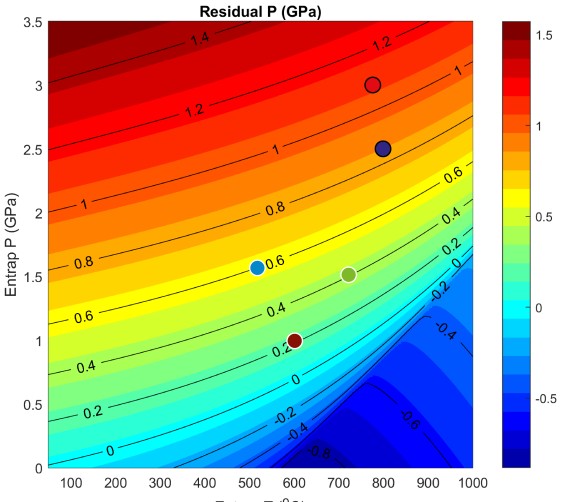 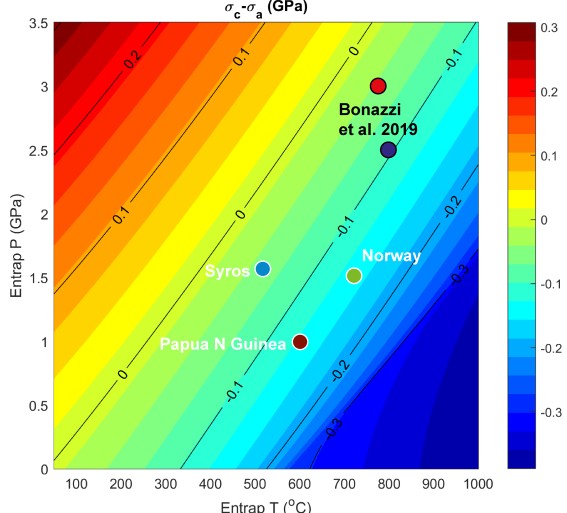


**Figure 2.** Model residual pressure and residual differential stress on quartz included in almandine. Expected residual pressure and stress corresponding to conditions of entrapment of experimental and natural inclusions are shown for comparison with measured values (Figs. 5 and 6). Actual pressures for natural inclusions may be shifted by less than 0.05 GPa due to compositional effects on garnet equation of state.


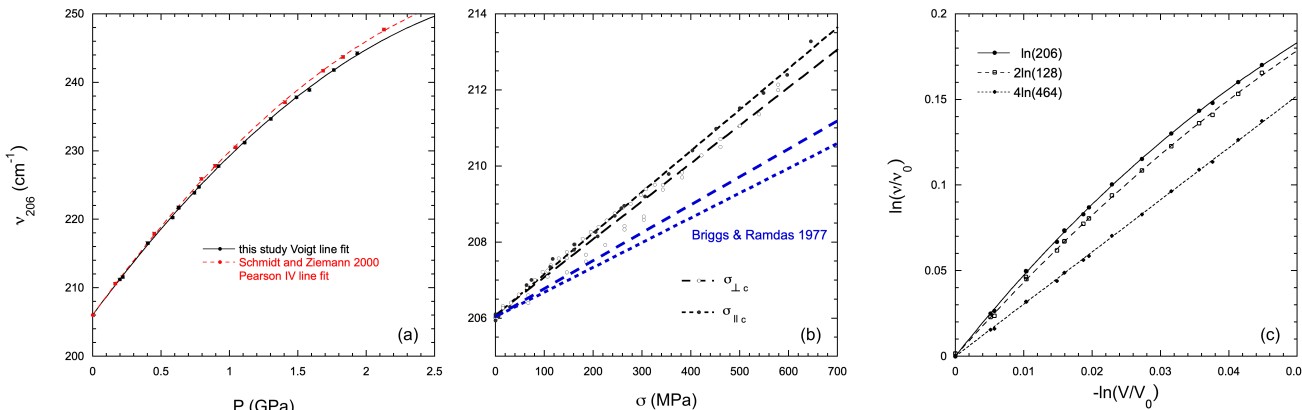

**Figure 3.** Dependence of the 206 cm$^{-1}$ peak on (a) hydrostatic pressure at ambient temperature and (b) uniaxial stress. The small discrepancy between present hydrostatic compression data and former experimental data (Schmidt and Ziemann, 2000)
is due to fitting with different peak shape (symmetrical Voigt used here instead of asymmetrical Pearson IV). Dependence on uniaxial stress obtained here are higher than those measured at 4 K (Briggs and Ramdas, 1977; Tekippe et al., 1973), due to temperature effects. The variations in peak position depend little on compression direction for the 206 cm$^{-1}$ peak in both studies, indicating its position depends essentially on hydrostatic pressure. (c) Grüneisen plot for hydrostatic compression showing constant Grüneisen parameter ($\gamma$) for the 464 cm$^{-1}$ peak and strong curvature for other peaks. Curves were fitted to
the data with a volume dependence of $\gamma$ expressed as the $q$ parameter (see text and Table 1). For the sake of easy comparison, relative frequency shifts are multiplied by 2 and 4 for the 128 and 464 cm$^{-1}$ peaks, respectively. Pressure was converted to volume using quartz equation of state (Angel et al., 1997).

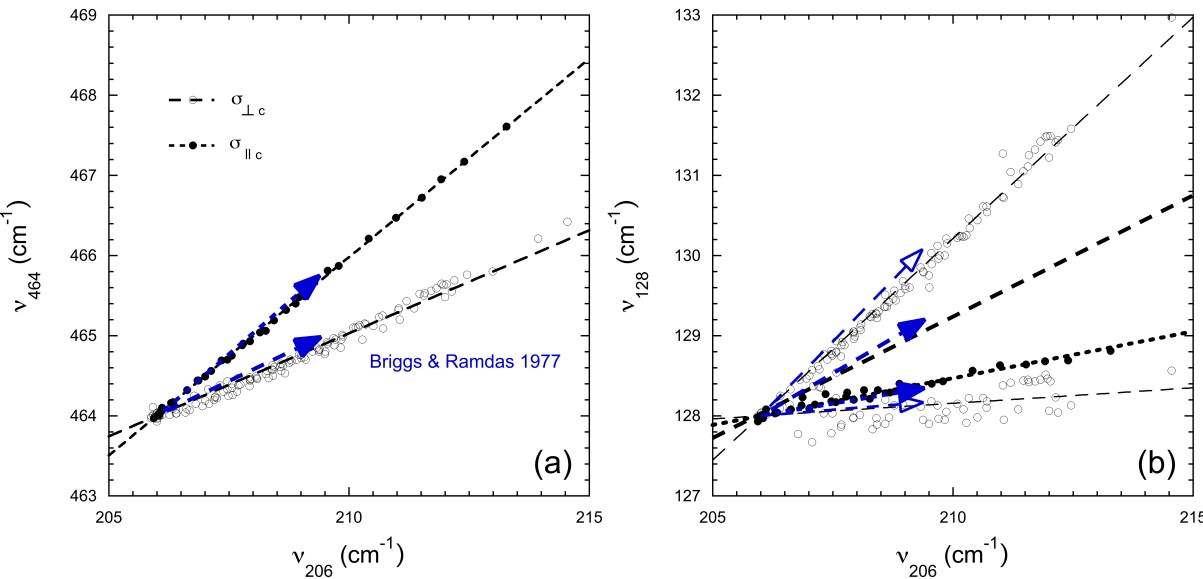


**Figure 4.** Stress dependence of the (a) 464 and (b) 128 cm$^{-1}$ peaks relative to the 206 cm$^{-1}$ peak at ambient temperature. Black lines show the relative shift from experiments at ambient conditions (this study), blue arrows the relative shifts for a uniaxial stress of 0.5 GPa at 4 K (Briggs and Ramdas, 1977; Tekippe et al., 1973). For the 128 cm$^{-1}$ peak, the two

components of the E mode are split by compression perpendicular to the c-axis (open circles and arrows). Stress dependence (long-dashed line) is the average of that of the two components (Briggs and Ramdas, 1977; Tekippe et al., 1973). A few data points were collected at effective stresses higher than 0.6 GPa in partially broken crystals, and the stress is then not calibrated because the crystal section is not known anymore. These points plot along extrapolation of the linear trends even though they were not included in the fit.


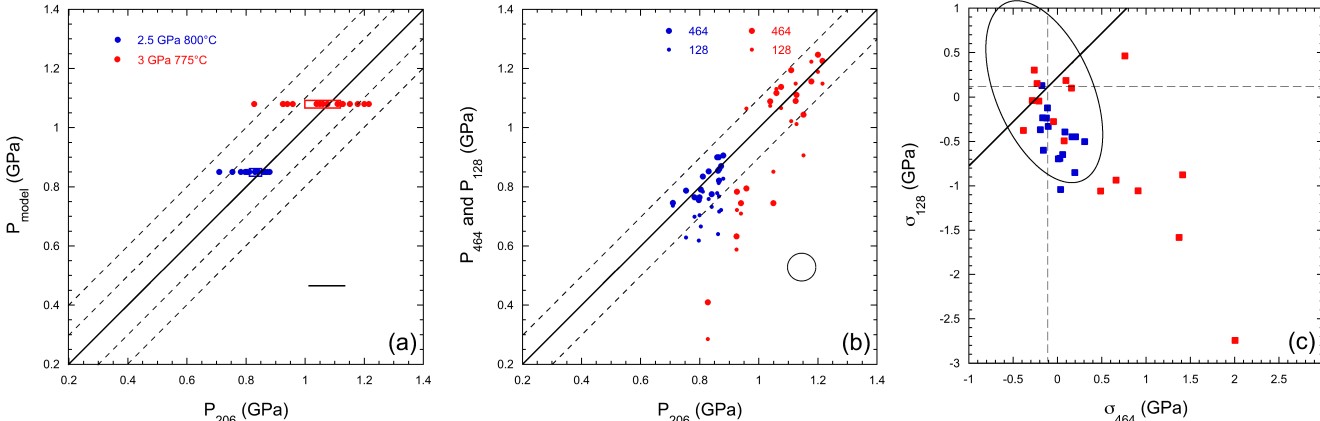

**Figure 5.** Pressures and stresses obtained from experimental quartz inclusions (Bonazzi et al., 2019). (a) Residual pressures from thermoelastic modeling (Fig. 2) as a function of pressures from the 206 cm$^{-1}$ peak for selected inclusions at the indicated conditions of formation. Empty rectangles show the mean value interval at 95% confidence level (2 standard error). Solid 1:1 line, dashed lines with 0.1 GPa offset are shown. Horizontal bar shows the uncertainty associated with 1 cm$^{-1}$ uncertainty on the 206 cm$^{-1}$ peak position. (b) Pressures from the 128 and 464 cm$^{-1}$ peaks as a function of pressure from the 206 cm$^{-1}$ peak. Residual pressures are in general agreement except for a group of experimental inclusions that show significant departure from the 1:1 line. Ellipse shows the uncertainties associated and 0.5 cm$^{-1}$ uncertainties on the 128-464 cm$^{-1}$ peak positions. (c) Residual differential stresses from the 128 and 464 cm$^{-1}$ peaks are consistent within uncertainties except for the group of inclusions departing from the 1:1 line by more than 1 GPa (see text). Ellipse shows the uncertainties associated with 1 and 0.5 cm$^{-1}$ uncertainties on the 206 and 128-464 cm$^{-1}$ peak positions, respectively. Anti-correlated uncertainties are associated with the 1 cm$^{-1}$ uncertainty on the 206 cm$^{-1}$ peak position that has opposite effects on the differential stresses calculated for the 128 and 464 cm$^{-1}$ peaks using equations (13) and (14), respectively (see text and supplementary Fig. S2).

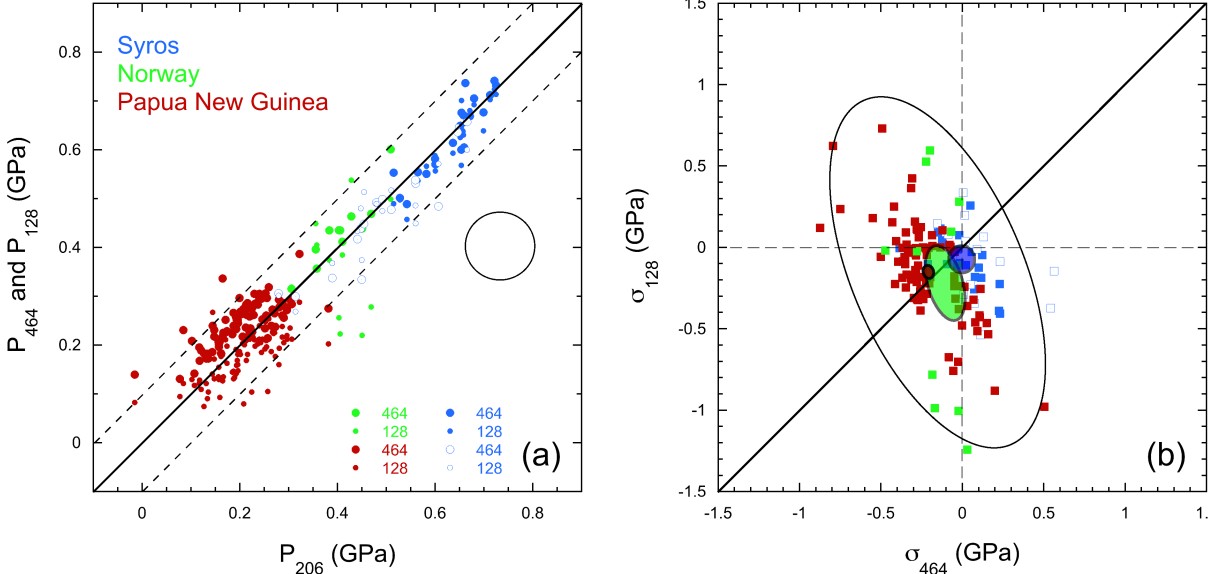

**Figure 6.** Stresses obtained from natural quartz inclusions. (a) Pressures from the 128 and 464 cm$^{-1}$ peaks deviate from pressure from the 206 cm$^{-1}$ peak with roughly opposite effects of similar magnitude in Papuas rocks, and differences are negligible in Syros blueschist. (b) Residual differential stresses from the 128 and 464 cm$^{-1}$ peaks. Mean values and confidence interval at 95 % are shown as colored ellipses. Dispersion of individual measurements are accounted for by systematic uncertainties on Raman measurements (black empty ellipse, same as in Fig. 5c). Residual stresses are negative in Papua-New Guinea and Norway rocks and close to zero in Syros blueschists. Data from Cisneros et al. (2021) for inclusions in garnet (filled symbols) and epidote (empty symbols) in blueschists from Syros (Cyclade islands, Greece), Gonzalez et al. (2019) for rocks from Papua-New Guinea, Zhong et al. (2019) for eclogitic inclusions from the Bergen Arcs (Norway). Eleven eclogitic inclusions from Norway were re-measured for this study with systematic comparison to reference spectrum. Large uncertainties on the 128 cm$^{-1}$ peak position on this dataset are associated with the high cutoff of the notch filter of the spectrometer in Berlin.

**Table 1.** Pressure and uniaxial stress dependence of the three major Raman peaks of quartz.

| | A | B | $\left(\frac{\partial v}{\partial P}\right)_0$ | $\gamma_0$ | q | $\left(\frac{\partial v}{\partial \sigma_1}\right)_{\sigma_3}$ | $\left(\frac{\partial v}{\partial \sigma_3}\right)_{\sigma_1}$ | $\gamma_1$ | $\gamma_3$ | $\gamma_1$ | $\gamma_3$ |
|---|---|---|---|---|---|---|---|---|---|---|---|
| | GPa/cm$^{-1}$ | GPa/cm$^{-2}$ | cm$^{-1}$/GPa | | | cm$^{-1}$/GPa | cm$^{-1}$/GPa | | | § | § |
| 128 | 0.127(6) | 0.0051(5) | 7.9(4) | 2.26(4) | 4.7(5) | 0.4(3)* | 1.2(1)** | 0.11 | 0.99 | *1.21* | *2.69* |
| 128* | | | | | | 5.5(3)* | | 4.14 | 2.02 | | |
| 206 | 0.0326(14) | 0.00046(4) | 30.7(13) | 5.04(6) | 6.4(3) | 10.0(4)* | 10.8(2)** | 5.18 | 6.71 | *3.64* | *5.25* |
| 464 | 0.107(3) | 0.0008(2) | 9.4(3) | 0.76(0) | -# | 2.6(2)* | 5.3(1)** | 0.66 | 1.34 | *0.60* | *1.19* |

§ First-principles results for trigonal strains from Murri et al. (2019) are shown in italics; * compression perpendicular to the c-axis, the 128 cm$^{-1}$ E mode is split due to symmetry reduction to monoclinic; ** compression parallel to c-axis; # q is not fitted for the 464 cm$^{-1}$ peak.