# Peer review of "Quartz under stress: Raman calibration and applications to geobarometry of metamorphic inclusions"

_EGUsphere, 2023_

## Community Comment (CC1)

**Grüneisen tensors of quartz**

A comment on 'Quartz under stress: Raman calibration and applications to geobarometry of metamorphic inclusions' by Reynard and Zhong, doi: 10.5194/egusphere-2023-100

*R J Angel. IGG, CNR, Padova. 15 March 2023*

**Motivation**

The manuscript by Reynard and Zhong describes measurements of the changes in the frequencies of Raman lines of quartz under hydrostatic pressure and also non-hydrostatic stress in two different experiments. These measurements are potentially valuable for the interpretation of the Raman shifts of quartz inclusions trapped inside other mineral hosts in terms of the stress state of the inclusion, from which the entrapment conditions of the inclusion can be constrained (e.g. Kohn 2014; Angel et al. 2017; Zhong et al. 2020; Gilio et al. 2021). It has previously been proposed that the Raman shifts of minerals under deviatoric stress can be interpreted in terms of the phonon-mode Grüneisen tensors (e.g. Angel et al. 2019), and that the values of the components of the tensors can be determined by conducting ab initio Hartree-Fock/ Density Functional Theory (HF/DFT) simulations of crystal structures under different stress/strain conditions (Murri et al. 2019; Stangarone et al. 2019; Musiyachenko et al. 2021).

Reynard and Zhong conclude from their work that the values of the components of the phonon-mode Grüneisen tensors of quartz obtained from HF/DFT are incorrect and further conclude that the use of these tensors to describe the shift in Raman frequencies under stress is not appropriate. In this comment I will clearly separate and discuss two distinct issues:

1) Whether or not the phonon-mode Grüneisen tensor is the correct physical description of the change in Raman frequencies of a crystalline solid under deviatoric stress, and what its known limitations are.

2) Whether or not the values of the components of the phonon-mode Grüneisen tensors determined by HF/DFT or by experiment agree, and if these values are correct.

**Phonon-mode Grüneisen tensors**

Phonon-mode Grüneisen tensors are simply the anisotropic generalisation of the concept of phonon-mode Grüneisen parameters which are well-established as the appropriate description of the relationship between the change in the wavenumbers of phonon modes with strain. Each phonon mode $m$ with a wavenumber $\omega^m$ is associated with a volume Grüneisen parameter $\gamma_V^m$ defined as:

$$\gamma_V^m = \frac{-V}{\omega^m} \frac{d\omega^m}{dV} \qquad (1)$$

Thus, the values of the volume phonon-mode Grüneisen parameters can be determined from an experiment in which wavenumbers of phonons are measured, for example by Raman spectroscopy, while the volume of the crystal is changed, for example in a high-pressure experiment or an experiment in which the temperature of the crystal is changed.

The anisotropic extension of equation (1) requires that instead of considering the volume strain $\frac{dV}{V}$, the change in the shape of a crystal must be considered. This is described by the strain tensor, $\varepsilon_{ij}$, which is a symmetric second rank tensor (Nye 1957). Because the wavenumber $\omega^m$ of a phonon mode is a scalar, the volume phonon-mode Grüneisen parameter must be replaced in (1) by a second-rank symmetric tensor, so that:

$$\frac{-d\omega^m}{\omega_0^m} = \boldsymbol{\gamma}^m : \boldsymbol{\varepsilon} \tag{2}$$

The ":" in Equation (2) indicates a double-scalar product between the two tensors, which can be written out in terms of their components as:

$$\frac{-d\omega^m}{\omega_0^m} = \gamma_{11}^m \varepsilon_{11} + \gamma_{22}^m \varepsilon_{22} + \gamma_{33}^m \varepsilon_{33} + +\gamma_{23}^m \varepsilon_{23} + \gamma_{32}^m \varepsilon_{32} + +\gamma_{13}^m \varepsilon_{13}$$
$$+ \gamma_{31}^m \varepsilon_{31} + \gamma_{12}^m \varepsilon_{12} + \gamma_{21}^m \varepsilon_{21} \tag{3}$$

Both tensors are symmetric (Nye 1957; Angel et al. 2019) and therefore $\varepsilon_{ij} = \varepsilon_{ji}$ and $\gamma_{ij}^m = \gamma_{ji}^m$ for each pair of non-diagonal elements, so:

$$\frac{-d\omega^m}{\omega_0^m} = \gamma_{11}^m \varepsilon_{11} + \gamma_{22}^m \varepsilon_{22} + \gamma_{33}^m \varepsilon_{33} + 2\gamma_{23}^m \varepsilon_{23} + 2\gamma_{13}^m \varepsilon_{13} + 2\gamma_{12}^m \varepsilon_{12} \tag{4}$$

We can reduce these tensors to a vector form in which the double-scalar product in Equation (2) becomes a scalar product of two vectors that represent the $\boldsymbol{\gamma}^m$ and the $\boldsymbol{\varepsilon}$ tensors. Under the Voigt convention, the normal strain components are equal in magnitude to the diagonal components of the tensor, e.g. $\varepsilon_1 = \varepsilon_{11}$, while the shear strains $\varepsilon_4, \varepsilon_5, \varepsilon_6$ are one-half of the values of the corresponding tensor components $\varepsilon_{23}, \varepsilon_{13}, \varepsilon_{12}$. Therefore, if we set $\gamma_4^m, \gamma_5^m$ and $\gamma_6^m$ equal to the values of the corresponding tensor components $\gamma_{23}, \gamma_{13}, \gamma_{12}$, we obtain an expression exactly equivalent to (4):

$$\frac{-d\omega^m}{\omega_0^m} = \gamma_1^m \varepsilon_1 + \gamma_2^m \varepsilon_2 + \gamma_3^m \varepsilon_3 + \gamma_4^m \varepsilon_4 + \gamma_5^m \varepsilon_5 + \gamma_6^m \varepsilon_6 \tag{5}$$

The introduction of a factor of ½ into the strain vector components and not into the Grüneisen vector components avoids factors of 2 appearing for the terms with subscripts $i = 4,5,6$ in the matrix version (5) of the tensor equation (2).

Because the phonon-mode Grüneisen tensors are properties of the crystal they are subject to the symmetry of the crystal. For this reason, the trigonal symmetry of quartz means that $\gamma_1^m = \gamma_2^m$ and $\gamma_4^m = \gamma_5^m = \gamma_6^m$ for each mode, so that each mode in quartz has only two unique non-zero components of its phonon-mode Grüneisen tensor, $\gamma_1^m \neq \gamma_3^m$. and thus equation (5) is reduced for quartz to:

$$\frac{-d\omega^m}{\omega_0^m} = \gamma_1^m(\varepsilon_1 + \varepsilon_2) + \gamma_3^m\varepsilon_3 \qquad (6)$$

Both the validation of this description for how the Raman frequencies of quartz change with stress or temperature, and the values of the components $\gamma_1^m$ and $\gamma_3^m$ for quartz can only be determined by experimental measurements or ab-initio simulations of quartz.

**Is the Grüneisen approach correct in principle?**

Both the isotropic (equation 1) and the anisotropic version (equations 3-5 in general, and 6 for quartz) predict that the changes in Raman mode shifts are linear with strain. For small strains, $\varepsilon_1 + \varepsilon_2 + \varepsilon_3 = \frac{dV}{V}$, so the isotropic and anisotropic approaches are consistent in this prediction. A further implication of this approach is that the phonon frequencies are a function of the strains of the crystal alone, and not of temperature or pressure. Therefore, if the Grüneisen approach is correct, a plot of any Raman frequency against the volume should lie on a single trend that is linear in volume, and this is observed for many modes of many crystals, including quartz (e.g. Murri et al. 2018).

[Figure]

Figure 1. Symbols show the measured changes in Raman shift under hydrostatic pressure from the experimental data of Morana et al. (2020) and Reynard and Zhong. The lines are the predicted shifts calculated from the measured unit-cell parameters of quartz under pressure (Scheidl et al. 2016) and the published Grüneisen tensor components of Murri et al. (2019) as solid lines and Reynard and Zhong (their Table 1) as dashed lines.

Note that the two sets of experimental data are very similar, as are the predicted shifts for the 128 and 464 lines which agree with the measured data up to ca. 2 GPa, a pressure greater than that found in quartz inclusions. The prediction of the 206 line from Reynard and Zhong does not describe the measured pressure evolution of this mode.

For small stresses, within the linear elastic regime where the strains are related to the applied stresses $\sigma_j$ (again in Voigt notation) by the elastic compliance matrix $s_{ij}$ as $\varepsilon_i = s_{ij}.\sigma_j$, the Raman shifts are predicted to change linearly with applied stress, as shown by Reynard and Zhong (their Fig 3c) and others (e.g. Tekippe et al. 1973; Briggs and Ramdas 1977; Barron et al. 1982). At high pressures, such as achieved in a DAC, the strains are no longer linearly proportional to the applied stress and instead the stress-strain relationship is described by non-linear equations of state. Thus, the Grüneisen approach implicitly predicts that Raman shifts will change non-linearly with pressure because of the non-linearity of strains with

pressure, even if the Grüneisen parameters or tensor components remain invariant with pressure. This is what is commonly observed in measurements of Raman shifts of crystals under hydrostatic pressure in diamond-anvil cells, including those of Reynard and Zhong (Fig. 1).

Further, for quartz and zircon, the mode Grüneisen parameters determined from HF/DFT simulations predict, in combination with the measured strains as a function of pressure, the experimentally measured changes in Raman frequencies with pressure (see my Fig. 1 for quartz). This agreement extends to volumes strains of ca. 4 % in both quartz (Murri et al. 2019) and zircon (Stangarone et al. 2019), corresponding to pressures of, respectively, 2 and 8 GPa. The mode Grüneisen parameters calculated for the 128 and 464 modes by Reynard and Zhong also show the same agreement (Fig. 1).

This brief discussion shows that there is substantial experimental evidence to indicate that the principles behind the Grüneisen approach are valid.

*Potential limitations of the Grüneisen approach*

We now consider possible limitations to the *extent* of its applicability, because if Grüneisen theory is found to be a valid description of the behaviour of minerals in the range of stresses and strains found in natural inclusions, as Figure 1 suggests, then it will be a useful tool in geology.

The first potential limitation is that it assumes a linear relationship between the phonon frequencies and the strains of the crystal. This may break down for several reasons. The first is already illustrated by Figure 1, that shows for pressures above ca. 2 GPa for quartz, the Raman shifts diverge from the prediction of the phonon-mode Grüneisen tensors. As noted above, this corresponds to about 4% in volume compression, and about 1.3% in linear strains. The same limiting strain value is found for zircon (Stangarone et al. 2019). This is not a breakdown of the Grüneisen approach, but simply requires an extension to non-linear relationships. This would be entirely analogous to the fact that the linear relationship between stress and strain breaks down at quite modest stresses (pressures) and the relationship must be described by non-linear EoS. But this is not a limitation to the method for interpreting Raman spectra of natural quartz inclusions as they exhibit pressures of less than 1.5 GPa.

There is also evidence from HF/DFT simulations (Murri et al. 2019) that at large strains some modes, and especially those involved in soft mode phase transitions such as the α−β transition in quartz, become non-linear in strain. Leaving aside other considerations, the physical properties of a mineral such as quartz change rapidly as the α−β transition is approached; for example the bulk modulus of quartz drops to zero at the transition (Lakshtanov et al. 2007) and other properties such as the heat capacity and thermal expansion coefficient diverge towards infinite values (e.g. Carpenter et al. 1998; Murri and Prencipe 2021). Given that the thermal expansion and heat capacity depend on the phonon frequencies, it would be entirely expected that the relationship between those frequencies and the cell parameters and strain may also become non-linear near to the phase transition.

The phonon-mode wavenumbers depend upon the interactions between the atoms within the crystal, which depend in part on the distances between them. One can associate the changes in phonon-mode wavenumbers with changes in the inter-atomic distances. For simple

structures, such as the rock salt structure in which all of the atoms have fixed coordinates within the unit cells, changes in inter-atomic distances are determined by the change in the unit-cell parameters alone. Therefore, there is good reason to expect a direct linear relationship between phonon-mode wavenumbers and unit-cell strains. However, in more complex structures such as quartz, the atom coordinates within the unit cell are independent variables. Therefore, there is no a-priori physical reason why the strains (fractional length changes) of inter-atomic distances or bonds should scale with the unit cell strains; indeed certain structural elements such as $SiO_4$ tetrahedra have much higher bulk moduli than the mineral structures that they form. Experimental determinations of bond lengths by diffraction methods is challenging because of both the small changes involved and the role of correlated thermal motion that prevents diffraction returning true local bond lengths (e.g. Busing and Levy 1964; Downs et al. 1992) especially in open framework structures such as quartz (Kihara 1990; 2001; Kimizuka et al. 2003; Murri et al. 2019). On the other hand, HF/DFT simulations do suggest that at low temperatures away from the α−β transition, that the real inter-atomic distances scale approximately linearly with the unit-cell strains (Murri et al. 2019). The same was found for zircon (Stangarone et al. 2019). Therefore, the internal degrees of freedom in mineral structures do not appear to create a significant limitation to the Grüneisen approach.

**The Grüneisen relationship in terms of stress**

For linear elasticity the Grüneisen relationships, both isotropic and anisotropic, can be written directly in terms of stress, by using the linear relationship between stress and strain, as Reynard and Zhong do. This is not wrong, but has two major disadvantages:

First: that it is not immediately obvious from the components of the stress tensor whether or not the symmetry of the crystal has been broken. This is important, because when the symmetry of the crystal is broken by the stress, then the forces and distances between the atoms are changed and the symmetries and frequencies of the phonon modes therefore change. Since the phonon-mode Grüneisen tensor is a property tensor and is therefore subject to the symmetry of the crystal (Eqn. 6) it only describes the changes in phonon-mode wavenumbers when the symmetry of the crystal is preserved. Therefore, a phonon-mode Grüneisen tensor of quartz calculated for trigonal symmetry is not expected to predict the wavenumber shifts of a quartz crystal whose symmetry has been broken by applied stress (Murri et al. 2022), and this is a problem in the analysis of Reynard and Zhong. They describe two non-hydrostatic stress experiments, one with stress applied along the c-axis of quartz, and the second with the stress applied perpendicular to the c-axis. In the first case, the only non-zero component of the stress vector is $\sigma_3$. For the second case Reynard and Zhong provide insufficient information about crystal orientations and the Cartesian axis convention used to determine the values of $\sigma_1$ and $\sigma_2$. What is however certain is that in this experiment the non-zero components of the stress field are some combination of $\sigma_1$ and $\sigma_2$, but they are not required to be equal (note this contradicts the statement made on lines 154-155 of Reynard and Zhong). In neither case is it obvious what is the symmetry of the quartz crystal under these stresses. If we convert these stresses into strains, then the answer is obvious. The first stress field induces strains $\varepsilon_1 = \varepsilon_2 \neq \varepsilon_3$ and the symmetry is preserved. The second, if the stress is just $\sigma_1 \neq 0$, $\sigma_2 = \sigma_3 = 0$, will produce a compressive strain along the Cartesian X axis, and expansion along the Cartesian Y and Z axes. Therefore, if we assume that the Cartesian X-axis is aligned along the crystallographic a-axis, the a-axis will be

shortened, while the crystallographic b and c parameters will be expanded. Unequal strains along the crystallographic a- and b-axes accompanied necessarily by a shear strain thus will result from all uniaxial stresses applied to quartz perpendicular to the c-axis. Therefore, the symmetry will be broken and the use of the trigonal phonon-mode Grüneisen tensors is not valid. Further, such a symmetry reduction does not result in LO-TO splitting of the E modes as claimed by Reynard and Zhong, but the generation of a pair of modes with different wavenumbers from the doubly-degenerate E modes of the parent trigonal structure. If the strained structure has monoclinic symmetry (as would occur for stress applied exactly along the crystallographic a-axis) the parent E mode splits into one A and one B mode. For other stress directions, the strained structure has triclinic symmetry and the E modes of the parent structure split into a pair of A modes (Tekippe et al. 1973; Murri et al. 2022). In both cases, each of these modes has an LO and TO component.

Note also that the LO-TO splitting in general of the E modes in quartz does not require the application of stress. The LO and TO components can be easily resolved in a well oriented crystal under certain scattering geometries, although the scattering geometry used by Reynard and Zhong is not specified. However, the E mode near 128 cm$^{-1}$ is the only one for which the LO-TO splitting cannot be resolved in unstressed quartz. Therefore, the evidence of splitting of this Raman peak into two different peaks requires symmetry breaking; the splitting of the 128 E mode reported by Reynard and Zhong is therefore not the TO-LO splitting of one parent E mode, but two different modes under the broken symmetry, neither of which can be expected to have properties predicted by the trigonal properties and Grüneisen tensors of quartz.

The second problem in using the applied stress as a basis for predicting changes in Raman or other phonon frequencies, as illustrated by the text of Reynard and Zhong, is that it is linear in the stresses (their equation 4, and equation 7 here). As I have stated above, apart from identifying symmetry breaking, under linear elasticity the relationship between mode wavenumbers written in terms of stress is entirely equivalent to that written in terms of strain. However, the changes in wavenumbers with large stress, as illustrated by the experiments under hydrostatic pressure, are non-linear (Fig. 1). If the wavenumber shifts are expressed in terms of pressure, then this requires two separate analyses for small stresses and large stresses, and the cross-over point from one approach to the other is not defined; indeed such an approach suggests that the behaviour of solids under small single stresses and under significant hydrostatic pressure is fundamentally different. This is not correct.  In contrast, the expression of wavenumber shifts in terms of strains has the great advantage that the non-linearity of wavenumber shift with high pressures emerges automatically and naturally not as a change in behaviour but simply as a consequence of the non-linearity of strains with pressure (Fig. 1).

Thus, we see that a description of an experiment, and the Grünesien relationship, in terms of strain leads to a clear identification of symmetry-breaking and thus the applicability or otherwise of the phonon-mode Grüneisen tensors. And it unifies the physical description of the response of the Raman modes to both small stresses and large hydrostatic pressure.

**Values of the components of the Grüneisen tensors**

Having established that there is sufficient experimental evidence that the Grüneisen theory of solids can predict the hydrostatic evolution of the Raman spectra of quartz (my Fig. 1, Murri

et al. 2019), zircon (Stangarone et al. 2019) and rutile (Musiyachenko et al. 2021) up to about 4% compression, we can now address the question as to how to determine the values of the components of the phonon-mode Grüneisen tensors.

Previous work (e.g. Tekippe et al. 1973; Briggs and Ramdas 1977) has used mechanical deformation, as Reynard and Zhong have done. This is a conceptually simple approach but has potential pitfalls. The most serious is that it is assumed that the stress applied to the load cell compressing the sample is applied completely to the sample (unless a friction correction is made), and that the sample is under a uniform homogeneous stress. The obvious problem is that, unless the sample is completely constrained in directions perpendicular to the load axis it will expand in the equatorial plane. If the stress and strain is homogeneous then this can be calculated from the elastic tensors of the crystal. However, an end-loaded sample will in general become barrel shaped; wider at the middle of the sample than at the ends due to friction in contact with the loaded pistons. Therefore, in order to interpret the measured Raman shifts from such an experiment it is necessary to either demonstrate that the stress and strain state is homogeneous across the complete sample, or to independently determine the strain or stress at the point of the Raman measurement (which could be done by simultaneous X-ray diffraction). Reynard and Zhong do not report such measurements, so the actual stress state of their samples could be different from that inferred from the pressure of the load cell.

That may be one contribution to the difference in the Grüneisen parameters determined in the experiments of Reynard and Zhong and those by HF/DFT (Murri et al. 2019), and may also account for the failure of the Grünesien parameters of Reynard and Zhong to predict their own hydrostatic measurements of the 206 mode (Figure 1 above).

[Figure]

Figure 2: Plot of measured changes in Raman shift under uniaxial stress along the c-axis from Reynard and Zhong. Dashed lines are the fits of Reynard and Zhong to their data.

The solid lines are calculated from the published Grüneisen tensor components of Murri et al. (2019) and the strains calculated from the applied stress and the elastic compliance tensor of quartz at room T (Lakshtanov et al. 2007). Other published measurements of the elastic tensor give very similar results.

Apart from this mode, there is general agreement in the prediction of the hydrostatic evolution of the 128 and 464 modes from both HF/DFT and the experiments of Reynard and Zhong (Figure 1). The agreement between the predictions of HF/DFT and the uniaxial stress measurements is poorer (Figure 2). This may be due inhomogeneities in the stress field in the sample during the experiments. But it may also indicate short-comings in the use of HF/DFT simulations, which is why determinations of the Grüneisen tensors by different methods is so important. Possible explanations for the predictions of HF/DFT being in error centre around the fact that the HF/DFT simulations are performed at 0 K (at the static limit), and the experiments are performed at 300K. Therefore, if there is a significant change in the dynamics of the quartz between 0 K and 300K, for example in the phonon-phonon interactions, then one would expect that the prediction of the Raman shifts at room temperature to be in error. However, then one would also expect that the predicted Raman shifts with pressure at room temperature (Figure 1) would be wrong, but they are actually correct. These discrepancies between mechanical experiments and predictions of HF/DFT can only be resolved by further careful evaluation and cross-comparison of both approaches.

The other point of agreement between the experiments of Reynard and Zhong and HF/DFT is in the sensitivity of the three Raman modes of quartz to non-hydrostatic stress. This can be evaluated from the Grüneisen tensors of quartz (Murri et al. 2019) by calculating the angle in the $\varepsilon_1$- $\varepsilon_3$ strain space between the lines of predicted constant mode wavenumber (the isoshift lines of Murri et al., 2019), and the line of strains expected under hydrostatic stress. The HF/DFT-predicted isoshift line (Murri et al. 2019) of the 206 mode lies at 91° (i.e. almost perpendicular) to the line of hydrostatically-induced strains, whereas the isoshift lines of the 128 and 464 modes are approximately 80 degrees from the hydrostatic line. The HF/DFT simulations therefore predict that the 206 mode is the least-sensitive to non-hydrostatic stresses, and this appears to be confirmed by the results of Reynard and Zhong.

**Summary**

It is clear that there is the possibility that the values of the components of the phonon-mode Grüneisen tensors determined by HF/DFT at 0 K are not exactly correct. They are however sufficiently correct that they reproduce the hydrostatic evolution of the Raman modes of quartz up to 2 GPa, the discrepancies being of the same order as the differences (which are unexplained by Reynard and Zhong) between the data of Reynard and Zhong and the previous determinations by Schmidt and Ziemann (2000) and Morana et al. (2020). The Grüneisen tensors also reproduce the expected inclusion pressures of synthetic inclusions from their measured Raman spectra (Bonazzi et al. 2019), and natural inclusions (e.g. Gilio et al. 2022), although I agree with Reynard and Zhong that some inclusions appear to have unreasonably high deviatoric stress calculated by this method. Whether this reflects experimental difficulties in measuring Raman spectra, the high correlation between the $\varepsilon_1$ and $\varepsilon_3$ strains determined from Raman shifts via the phonon-mode Grüneisen tensors, or actually problems with the values of the tensor components determined by HF/DFT, remains to be determined.

Possible reasons for different values of the components of the phonon-mode Grüneisen tensors may lie either in short-comings of applying HF/DFT simulations to experiments at room temperature, or to short-comings in the deformation experiments of quartz. The reasons can only be determined through carefully documented experiments in which the stress or strain state of the sample at the point of Raman measurement is determined. Such experiments can only provide a test of the HF/DFT simulations if they are performed in

orientations and stress states that preserve the symmetry of the quartz crystal. Thus, the second uniaxial stress experiment of Reynard and Zhong with the stress applied perpendicular to the c-axis broke the crystal symmetry of quartz (as evidenced by the splitting of the mode into two modes, not the LO and TO components of one mode), and cannot be used for comparison to the predictions from the HF/DFT simulations of trigonal quartz. Further experiments are clearly required before any conclusions can be drawn about the correctness or otherwise of the component values of Grüneisen tensors calculated by HF/DFT. In any case, whether or not the values are correct, no evidence has yet emerged to disprove the general concept of Grüneisen that the phonon-mode wavenumbers scale with the strains applied to the crystal.

**References**

Angel, R.J., Mazzucchelli, M.L., Alvaro, M., and Nestola, F. (2017) EosFit-Pinc: A simple GUI for host-inclusion elastic thermobarometry. American Mineralogist, 102, 1957-1960.

Angel, R.J., Murri, M., Mihailova, B., and Alvaro, M. (2019) Stress, strain and Raman shifts. Zeitschrift für Kristallographie, 234, 129-140.

Barron, T.H.K., Collins, J.F., Smith, T.W., and White, G.K. (1982) Thermal expansion, Grüneisen functions and static lattice properties of quartz. Journal of Physics C: Solid State Physics, 15, 4311-4326.

Bonazzi, M., Tumiati, S., Thomas, J., Angel, R.J., and Alvaro, M. (2019) Assessment of the reliability of elastic geobarometry with quartz inclusions. Lithos, 350-351, 105201.

Briggs, R.J., and Ramdas, A.K. (1977) Piezospectroscopy of the Raman spectrum of $\alpha$-quartz. Physical Review B, 16, 3815-3826.

Busing, W.L., and Levy, H.A. (1964) The effect of thermal motion on the estimation of bond lengths from diffraction measurements. Acta Crystallographica, 17, 142-146.

Carpenter, M.A., Salje, E.K.H., Graeme-Barber, A., Wruck, B., Dove, M.T., and Knight, K.S. (1998) Calibration of excess thermodynamic properties and elastic constant variations associated with the alpha-beta phase transition in quartz. American Mineralogist, 83, 2-22.

Downs, R.T., Gibbs, G., Bartelmehs, K.L., and Boisen, M.B. (1992) Variations of bond lengths and volumes of silicate tetrahedra with temperature. American Mineralogist, 77, 751-757.

Gilio, M., Angel, R.J., and Alvaro, M. (2021) Elastic geobarometry: how to work with residual inclusion strains and pressures. American Mineralogist, 106, 1530-1533.

Gilio, M., Scambelluri, M., Angel, R.J., and Alvaro, M. (2022) The contribution of elastic geobarometry to the debate on HP versus UHP metamorphism. Journal of Metamorphic Geology, 40, 229-242.

Kihara, K. (1990) A X-ray study of the temperature dependence of the quartz structure. European Journal of Mineralogy, 2, 63-77.

--- (2001) Molecular dynamics interpretation of structural changes in quartz. Physics and Chemistry of Minerals, 28, 365-376.

Kimizuka, H., Kaburaki, H., and Kogure, Y. (2003) Molecular-dynamics study of the high-temperature elasticity of quartz above the $\alpha-\beta$ phase transition. Physical Review B, 67, 024105.

Kohn, M.J. (2014) "Thermoba-Raman-try": Calibration of spectroscopic barometers and thermometers for mineral inclusions. Earth and Planetary Science Letters, 388, 187-196.

Lakshtanov, D.L., Sinogeilin, S.V., and Bass, J.D. (2007) High-temperature phase transitions and elasticity of silica polymorphs. Physics and Chemistry of Minerals, 34, 11-22.

Morana, M., Mihailova, B., Angel, R.J., and Alvaro, M. (2020) Quartz metastability at high pressure: what new can we learn from polarized Raman spectroscopy? Physics and Chemistry of Minerals, 47, 34.

Murri, M., and Prencipe, M. (2021) Anharmonic Effects on the Thermodynamic Properties of Quartz from First Principles Calculations. Entropy, 23, 1366.

Murri, M., Alvaro, M., Angel, R.J., Prencipe, M., and Mihailova, B.D. (2019) The effects of non-hydrostatic stress on the structure and properties of alpha-quartz. Physics and Chemistry of Minerals, 46, 487-499.

Murri, M., Gonzalez, J.P., Mazzucchelli, M.L., Prencipe, M., Mihailova, B., Angel, R.J., and Alvaro, M. (2022) The role of symmetry-breaking strains on quartz inclusions in anisotropic hosts: implications for Raman elastic geobarometry Lithos, 422-423, 106716.

Murri, M., Mazzucchelli, M.L., Campomenosi, N., Korsakov, A.V., Prencipe, M., Mihailova, B., Scambelluri, M., Angel, R.J., and Alvaro, M. (2018) Raman elastic geobarometry for anisotropic mineral inclusions. American Mineralogist, 103, 1869-1872.

Musiyachenko, K.A., Murri, M., Prencipe, M., Angel, R.J., and Alvaro, M. (2021) A new Grüneisen tensor for rutile and its application to host-inclusion systems. American Mineralogist, 106, 1586.

Nye, J.F. (1957) Physical properties of crystals. Oxford University Press, Oxford, p 329.

Scheidl, K., Kurnosov, A., Trots, D.M., Boffa-Ballaran, T., Angel, R.J., and Miletich, R. (2016) Extending the single-crystal quartz pressure gauge to hydrostatic pressures of 19 GPa. Journal of Applied Crystallography, 49, 2129-2137.

Schmidt, C., and Ziemann, M.A. (2000) In-situ Raman spectroscopy of quartz: A pressure sensor for hydrothermal diamond-anvil cell experiments at elevated temperatures. American Mineralogist, 85, 1725-1734.

Stangarone, C., Angel, R., Prencipe, M., Campomenosi, N., Mihailova, B.D., and Alvaro, M. (2019) Measurement of strains in zircon inclusions by Raman spectroscopy. European Journal of Mineralogy, 31, 685-694.

Tekippe, V.J., Ramdas, A.K., and Rodriguez, S. (1973) Piezospectroscopic Study of the Raman Spectrum of α-Quartz. Physical Review B, 8, 706-716.

Zhong, X., Moulas, E., and Tajčmanová, L. (2020) Post-entrapment modification of residual inclusion pressure and its implications for Raman elastic thermobarometry. Solid Earth, 11, 223-240.

---

## Author Comment (AC2)

**We thank the referee for the detailed and insightful comments, which we have addressed as noted in bold in the following. Comments are addressed in the text as highlighted in yellow.**

*Review of Reynard and Zhong: "Quartz under stress: Raman calibration and applications to geobarometry of metamorphic inclusions"*

Dear Editor,

I have read through this manuscript in detail. It presents a new experimental calibration of the three major Raman peaks of quartz with hydrostatic pressure and uniaxial differential stress. I believe this is a valuable contribute to the discussion on the variations of Raman shifts of quartz under an applied stress and strain. In this respect the experiments reported in this manuscript bring a useful contribute to the development of a methodology to assess the stress states of inclusions in their host minerals for mechanical thermobarometry applications. The possibility to use the differential stress of several inclusions in the same host to obtain independent constraints on the temperature of entrapment, even if suggested already in previous literature, is appealing. For this reason, a robust method to quantify the differential stress of inclusions with Raman spectroscopy would be extremely important in the field of petrology. However, I have some concerns about this manuscript which should be addressed by the authors. I believe that this work is suitable for publication in Solid Earth, provided that the following comments are addressed.

I introduce my general comments here, and further details are presented in the specific comments below.

General comments

The calibrations presented in this manuscript are based on the results from Raman experiments of quartz under hydrostatic pressure and uniaxial stresses with different orientations. In one of these experiments, the authors observe a separation of the doubly degenerate (E) mode (near 128cm$^{-1}$) when the uniaxial stress is applied along the *a*-axis of the quartz unit-cell. This behavior is identified by the authors as the LO-TO splitting of the E mode, and they conclude that their results are directly applicable to quartz crystals with trigonal symmetry. However, their interpretation does not agree with the existing Raman theory and the requirements of the symmetry analysis, as shown also by previous experiments and clearly summarized in its formal aspects by Tekippe et al. (1973). Depending on the orientation of the stress with respect to the unstrained trigonal unit-cell, this configuration leads to a reduction of symmetry from trigonal to monoclinic or triclinic. Therefore, the splitting observed in the experiments of Reynard and Zhong is not the LO-TO splitting of the E mode, but rather the splitting of the E mode into totally symmetric (A) and antisymmetric (B) non-degenerate modes (monoclinic symmetry), or into a pair of A modes (triclinic symmetry). Indeed, E modes in monoclinic and triclinic crystals are not allowed by symmetry. This is a crucial point, since the authors refer to the previous literature (Tekippe et al., 1973; Murri et al., 2022) to support the argument that their experiments apply directly to trigonal quartz inclusions without symmetry breaking, while those works conclude the opposite. Since it is possible that quartz inclusions in synthetic and natural samples have symmetry lower than trigonal because of their stress state, the results of calibrations with symmetry breakings are still useful and should be considered for the characterization of the stress states of inclusions in rocks. For this reason, I recommend that the discussion is revised in order to make it consistent with the widely known physical theory, and to discuss the implications of the symmetry breaking. I elaborate further on this point in the specific comments below.

**We agree with the referee that the phrasing "TO-LO" splitting is incorrect because the splitting of the two components of the 128 E mode is due to symmetry reduction for compression at non-zero angle to the *c*-axis. We have replaced it by the expressions "splitting of the 128 mode components" or "splitting of the 128 peak" throughout the text. We respond to other specific comments on symmetry reduction below.**

Moreover, as a potential user of the calibrations proposed in this manuscript, I was confused to realize that two different calibrations are proposed to estimate the same quantity (i.e. the differential stress in an inclusion), but often the values obtained from them do not agree. The disagreement arises from an apparent anti-correlation of the two proposed calibrations, the reasons for which are unfortunately not discussed in the manuscript. The averaging of the results of the two calibrations over many inclusions is proposed as a procedure to obtain the estimate of the residual differential stress of the sample, but even after this averaging the results often do not agree within uncertainties. I believe that, in order to improve the reception of the proposed approach by the scientific community, it would be important to expand the discussion on the reason why several averaging steps of the results of two calibrations that appear anti-correlated make this a robust method to estimate the residual differential stress. I elaborate more on this point in the specific comments below.

**For quartz inclusions in garnet, there are two independent variables, hydrostatic pressure P and differential stress $\sigma$, and we have two independent equations to solve it. P is determined from the shift of the 206 line that is independent of $\sigma$ and two independent values of $\sigma$ are determined from the shifts of the 128 and 464 lines. If other weaker lines were to be calibrated, we could define one independent estimate of $\sigma$ for each of them. Anticorrelation arises from uncertainties on P from the 206 line position and their propagation into the equations. Since this was not clear enough, we have expanded paragraphs to explain it as explained below, and added a supplementary figure to explain the error propagation.**

**Specific comments**

- Lines 52-53. In Fig. 1 the Ne (and not He) and Hg peaks are marked with different wavenumbers to what reported in the text.

- **The typo on Ne is now corrected. The absolute positions of the sharp emission bands were given in the text. The "zero" used to define the Raman shift at absolute position is 18786.05 cm-1 corresponding to the 532.31 nm wavelength of the laser. The difference between this value and the emission line position gives their apparent shift on Fig. 1 of 269.46 and 473.6 cm-1. Small shifts with respect to this value are due to intrinsic uncertainties on the zero position. Explanation is now given in line 54.**

- Lines 73-76. Since the details of the experiments under uniaxial stress affect the resulting calibration, I suggest that the authors add here more details. I suggest that they describe here the dimensions of the sample and the procedure they used to orient the crystals. Reynard and Zhong mention that they oriented the sample optically and refer to Tekippe et al. (1973), who however oriented their crystal with X-rays. Determining the precise orientation of the crystal is fundamental for the correct estimate of the effect of stress, also in the light of the symmetry breaking discussed in my comment above. A slight deviation in the orientation (i.e. a stress not exactly oriented along the *a*-axis) can lead to a symmetry reduction to triclinic rather than monoclinic. The authors should give more details on the procedure they used and how the estimated accuracy of 3° can be achieved with optical observations. If possible, it would also be useful to add a figure that shows the experimental setup.

- **Faces of parallelepipeds were first cut perpendicular to the c-axis of a large single-crystal. Orientation of basal cuts was confirmed under the polarizing microscope using conoscopic figures as described in Briggs and Ramdas (1977). Orientation was also checked by Raman spectroscopy on unused rods. The c-axis orientation was within 1°, and that of the perpendicular to the *a*-axis within 2° (in the the basal plane, rods were cut parallel and perpendicular to hexagonal faces of the crystal, i. e. either parallel to *a* or perpendicular to it). This does not affect the measured shifts since the same shifts and splittings of E mode components are expected along those two directions (Tekippe et al., 1973). Details were added lines 78-86, and figures of Raman orientation are given in supplementary material (Fig. S1) shown below.**

[Figure]

**Figure S1. Orientation of crystal determined by relative Raman intensities. Parallel and perpendicular cut are defined with respect to the *c*-axis. Orientation of the c-axis (top) and the perpendicular to the a-axis (bottom) is within 2° of the normal to the compression face (vertical direction).**

- Lines 80-87. I suggest that here a few more details are given regarding the elastic properties they used to calculate the strains of the quartz inclusions. They mention the EoS, but it is not clear if they used axial EoS (e.g. Angel et al., 2021) or some extrapolation of the elastic tensors to high pressure and temperature. They should also mention the source of the elastic properties of almandine that were used in the calculation, and if they are assumed to be constant with P and T or not.

- **Details and references were added in section 3 lines 91-94 and 96-99.**

- Line 116. The authors should precisely report the orientation of the applied stress with respect to the quartz crystal. Because of the symmetry of quartz, the *a-b* crystallographic plane is not elastically isotropic (Nye, 1985). Assuming that the Cartesian *x*-axis is aligned to the *a*-axis of the unit-cell, and the Cartesian *z* to *c*, a uniaxial stress applied along *x* or along a direction orthogonal to it, would reduce the symmetry to monoclinic. On the other hand, if the non-zero component of the uniaxial stress is applied in any other direction of the Cartesian *x-y* plane, the symmetry would be reduced to triclinic rather than monoclinic. Therefore, it is not enough to state that the stress is applied normal to the crystallographic *c*-axis, and the authors should state explicitly if the *a*-axis is parallel to the global Cartesian *x* and if they oriented the stress exactly along the *a*-axis.

- **As stated above, there is a deviation of the *c*- or *a*-axis of <2°. Perpendicular to c-axis, the symmetry reduction to monoclinic cannot be avoided in the uniaxial compression setup used here and in earlier studies. With the deviation of crystallographic axes from face normal, the triclinic distortion is negligible. This is explained in lines 133-136.**

- Line 117. Before discussing the evolution of Raman shift with the applied stress, I believe it would be worthwhile to discuss if any barreling effect does take place in the sample due to the friction with the compression cell, and if the authors assessed the homogeneity of the stress in the sample. The potential inhomogeneity of stress in the sample would require a correction to determine the exact value of stress at the point of the Raman measurements, given the stress applied to the crystal rod.

- **Raman shift were measured at half length of the crystal at a depth of about 200 micrometers. Barreling could indeed be a concern, although it would remain limited in**

**quartz with a maximal stress of 0.6 GPa. If we multiply this stress tensor (0, 0, 0.6) by the compliance tensor of quartz, we get 0.0007 lateral strain. This means the area correction is 0.0007^2, which is 5e-7 or ~300 Pa. Tekippe et al (1973) and Briggs and Ramdas (1979) do not mention issues with barreling for aspect ratio of ~3. Barreling should have more effect on short crystals than on long ones. This is not the case because we have measured the Raman shift using samples with different aspect ratios of 2.5 and 4 perpendicular to the c-axis, and 1 and 4 along the c-axis, and we obtained similar results. We added those details in lines 140-141 and crystal size data in the supplementary table.**

- Lines 126-132: as introduced in the general comments above, I have serious concerns regarding the interpretation of the splitting of the doubly degenerate (E) mode (near 128cm$^{-1}$) under uniaxial stress applied along the *a*-axis as a LO-TO splitting. The symmetry of a crystal under uniaxial compression is determined by the symmetry elements common to both the unstrained crystal and the strain state. The reduction of the symmetry of the crystal point-group as a result of an applied external field is known in the literature as the *morphic effect* (Anastassakis, 1980; Gregora, 2013). Assuming that the orientation of the stress in this experiment is exactly along the *a*-axis (which in turn is parallel to the Cartesian x) of the unstrained trigonal unit-cell of quartz, this configuration leads to a reduction of symmetry from trigonal to monoclinic (see table II of Tekippe et al., 1973). Therefore, in this configuration the E mode, belonging to the parent trigonal symmetry, splits into a totally symmetric (A) and antisymmetric (B) non-degenerate mode in the new stress-induced monoclinic symmetry, as required by the theory and already shown in previous experiments (Fig. 5, table II and discussion at page 813 in Tekippe et al., 1973). This should not be confused with the LO-TO splitting of the E mode, which occurs in non-centrosymmetric crystals (even unstressed) because of the long-range polarization fields (electro-optic effect; Gregora, 2013) and it is typically unresolved for the 128cm$^{-1}$ line (Briggs & Ramdas, 1977). A similar behavior was also confirmed by the first principle calculations of Murri et al. (2022) for a stress configuration which leads to a symmetry reduction from trigonal to triclinic. They showed that each E mode of an unstressed trigonal quartz, splits into two A modes because of the applied strain/stress that induces a symmetry breaking, and both of the A modes in turn exhibit a LO-TO splitting. Therefore, the separation of the 128cm$^{-1}$ peak observed by Reynard and Zhong with a uniaxial stress parallel to the *a-b* plane of quartz is actually the evolution of the E mode into an A and B modes (if the uniaxial stress is applied along the *a*-axis or orthogonal to it, leading to a monoclinic symmetry) or into two A modes (if the uniaxial stress is applied along any other direction in the *a-b* plane and the final symmetry is triclinic). In order to correctly interpret which of the situations takes place in these experiments (i.e. symmetry reduction to monoclinic or triclinic), Reynard and Zhong should clearly state the orientation of the Cartesian axes with respect to the crystallographic axes of the quartz sample, and the orientation of the stress in the Cartesian reference system (see also my previous comment). Moreover, the authors use the DFT calculations of Murri et al. (2022) to support that the results of this study are applicable to quartz inclusions with trigonal symmetry (lines 130-131). However, the conclusions of Murri et al. (2022) say the opposite. They state that when the splitting between two A(LO-TO) modes in the stress-induced triclinic symmetry (originated from one E(TO-LO) mode in the parent trigonal symmetry) becomes detectable, the use of the inclusion is not recommended for elastic thermobarometry due to the symmetry breaking. Therefore Reynard and Zhong should correct the wrong identification of this behavior as a LO-TO splitting of the E mode throughout this manuscript, in table 1 and in the figures. They should also not use the results from first principle calculations to support that the effect of symmetry reduction in their experiments is small and that their results are directly applicable to trigonal crystals. Since it is possible that quartz inclusions in synthetic and natural samples have symmetry less than trigonal because of their stress state, the results of calibrations with symmetry breakings still provide useful information for the characterization of the stress states of inclusions, and in my opinion the discussion in the manuscript should head towards this direction.

- **As stated above, we agree with the referee and the incorrect use of TO-LO splitting for symmetry-induced splitting was corrected throughout the text. In the reference Murri et al. (2022), it is stated in their abstract "***These HF/DFT simulations show that the changes in the positions of the Raman modes produced by strains that are expected for symmetry broken quartz inclusions in zircon are generally similar to those that would be seen if the quartz inclusions***

*remained truly trigonal in symmetry.***", which we interpret as symmetry breaking being of second order importance. Citation is now placed in the discussion of natural inclusions (line 297). An essential point is that the sum of individual stress dependences obtained here, even in symmetry breaking geometry, are similar to the hydrostatic shift at ambient pressure, showing that the effect of symmetry breaking, if any, is of second order. A paragraph was modified in lines 147-154 to clarify this point.**

- Lines 135-142. Since the authors compare their experimental results at hydrostatic conditions with the previous literature, they should also compare them with the hydrostatic results of Morana et al. (2020).

- **Reference to Morana et al. (2020) was already given in line 111 along with Hemley 1987, where we explained that data above 2 GPa yielded too low pressure dependences when compared with the present calibration. We have expanded a bit this paragraph (lines 125-127) to explain this source of uncertainty and how we remedy it using small pressure steps in the 2 GPa pressure range.**

- Line 172. This sentence seems to say that DFT calculations cannot be used as a basis to calculate the LO-TO splitting. Actually DFT calculations can be used to calculate the LO and TO components and therefore their splitting (see for example Murri et al., 2019). However, as discussed in previous comments, the experiments under uniaxial stress in the *a-b* plane presented by Reynard and Zhong lead to a symmetry reduction to monoclinic (or triclinic) which is different to the DFT results reported in Murri et al. (2019) where no symmetry breaking occurred and the trigonal symmetry was preserved in all of the simulations. This might potentially contribute to part of the discrepancy between the results presented here and the Grueneisen tensor of Murri et al., (2019).

- **We thank the referee for pointing out this mistake of calling it LO-TO splitting and clarifying what was calculated by Murri et al. (2019). We agree that the symmetry breaking may have a second order contribution to the discrepancy as discussed above (line 152-154). We also emphasize that non-linear variations of Grüneisen parameters also contribute to discrepancies between fits to first-principles calculation and experiments (lines 170-172).**

- Sections 6.1 and 6.2. I approached these sections as a possible user of these calibrations, and I am a bit confused by the trends shown in the reported results. In this manuscript two different calibrations (sigma128 and sigma 464) are proposed to estimate the same quantity, the value of differential stress in one inclusion. One should expect that these calibrations give the same (or similar) results when applied to the same inclusion. In other words, the stress determined in this way should fall on the 1:1 line in Fig. 5C and 6B. However, one can see that the stress of inclusions belonging to the same dataset always align along a line that forms a high angle to the 1:1 line. Therefore, often, the two calibrations give very different values of differential stress for one inclusion, which may have even opposite sign (see Fig. 5C and 6B). The fact that the inclusions in all samples show this anti-correlation between the values obtained from the sigma128 and sigma464 calibrations, points to the fact that this is a feature arising from the calibration and not from the specific features of a sample, because clearly inclusions cannot have simultaneously two different states of stress. The authors propose arbitrarily that inclusions are "valid" if the values of (sigma3-sigma1) from the two calibrations differ by less than 1 GPa (lines 222-223). The reason for choosing such a large threshold value is not discussed in the manuscript, but this choice means that the proposed methodology considers acceptable that one inclusion is simultaneously under a (sigma3-sigma1) = -0.49 GPa and (sigma3-sigma1) = +0.49 GPa. In this respect, the statement that the results of the two calibrations are self-consistent (lines 251-261) is very misleading (see also my specific comment below, with some statistics on the given results). Such results may be confusing for a user. Clearly an inclusion cannot be simultaneously under two different stress states, and therefore this method does not determine the differential stress of one inclusion (except in rare cases). In my opinion, the reason for the anti-correlation between the values obtained from the sigma128 and sigma464 calibrations should be explained, because it is not intuitive and it is not clear by which physical behavior it is originated. Moreover, when many inclusions in one sample are measured, the results of the sigma128 and sigma464 calibrations

averaged on all the inclusions in the sample give somewhat similar results, but they often do not agree within mutual uncertainties (see the datasets of Cisneros et al., 2020 and Gonzalez et al., 2019 in the supplementary table of Reynard and Zhong). The authors therefore suggest that the averaged results of the two calibrations should be averaged again to give the final correct value. By reading the manuscript it is not clear to me the reason why several averaging steps of the results of two calibrations that appear anti-correlated should make this a robust method to estimate the value of differential stress. I had the (maybe wrong) impression that the motivation is just the fact that the final averaged value is somewhat similar to the expectations of the elastic model for some of the analyzed samples. Therefore, if the authors have more physical insights that support the robustness of this procedure I suggest that they discuss them in order to avoid possible confusion for the reader.

- **We agree that confusion must arise from insufficient explanation of the way we treat the data and why. ==We have expanded the discussion line 232-252 to provide more details.== For each inclusion, the pressure is calculated from the 206 line position that is independent of differential stress using equation (1). Two estimates of the differential stress are then obtained using equations (13,14). As Raman measurements are subject to statistical fluctuations due to uncertainties, the two stresses are likely to differ for a given inclusion. Uncertainty in pressure will arise from uncertainty in the position of the 206 peak. Uncertainty on stress will arise from uncertainty in the position of the 126 and 464 peaks, and also in the position of the 206 peak that is used to calculate pressure in eq (13,14). Based on eq. 13 and 14, differential stress is linearly correlated to the 128 and 464 Raman peak position, thus any uncertainties arising solely from Raman measurement will directly propagate into the calculated differential stress. Also, because the 128 and 464 peak shifts induced by stress are of opposite sign, a fluctuation of the 206 peak position will result in an apparent anticorrelation between stresses obtained using eq (13,14), as seen in Figures 5 and 6. As the differential stresses calculated from equations (13,14) are in general small, they are very sensitive to small systematic errors in Raman measurements, as shown on the new supplementary figure S2 reproduced below.**

[Figure]

**Fig. S2.** Sensitivity analysis for calculated differential stresses $\sigma$ using the position of the 128 and 464 cm$^{-1}$ peaks (equations 13,14). Raman band position is calculated using Eq. 2 and the inverted Eq. 13 and 14 at a given stress state, here set as P = 1GPa and $\sigma$ = -0.1GPa. Subsequently a Gaussian noise is added to the calculated Raman band position, and Eq. 1, 13 and 14 are again used to calculate differential stress. This is repeated 10000 times. Different variances in Raman band position are also tested to systematically show the sensitivity. Calculated stresses are extremely sensitive to small standard deviations on measured Raman frequencies. Anti-correlation increases as the standard deviation on the 206 cm$^{-1}$ peak gets

higher (top row), as seen in Figs. 5c and 6c for actual measurements on experimental and natural inclusions. Assuming conservative values of standard deviation of 1 cm$^{-1}$ on the 206 cm$^{-1}$ peak, and of 0.5 cm$^{-1}$ on the other peaks (top right diagram), absolute differences between the two values of $\sigma$ higher than ~1 GPa are deemed unlikely and such values are rejected.

If a sufficient number of inclusions is measured, and if they belong to the same population (i.e. if they formed in a single event under similar pressure and temperature conditions), the average pressure and stress values can be obtained with a greater accuracy since the standard error of the mean will be greatly reduced. An alternative approach would be to use the overdetermination of the system (2 unknowns and 3 independent equations or more if more peaks are used) to obtain the best-fit values of pressure and differential stress or strain (Bonazzi et al 2019, Murri et al 2019). It is our choice to compare the stresses obtained from eq (13,14) because it allows checking the self-consistency of the Raman measurements that may other wise be blurred if a single best fit value is calculated for each inclusions.

- Lines 229-230. According to Bonazzi et al., (2019) they applied the Grueneisen tensor components of Murri et al. (2019) to calculate the strains from Raman shifts, and then they used the room-pressure elastic tensor of Wang et al. (2015) to calculate the stresses from strains, without further corrections. I am not sure why the authors say that the relationship between strain and stress is not provided in Bonazzi et al., (2019), and to which "correction" they refer. This sentence should be explained in more details or removed.

- **We thank the referee for the clarification, the sentence was removed.**

- Lines 235-236. Apparently the dataset Alm-1 (synthesis at 3GPa, 775°) of Bonazzi et al. (2019) is completely discarded, since it cannot be explained by the proposed calibrations. The authors state that those measurements were affected by "systematic uncertainties in the Raman peak positions". However, Reynard and Zhong neglect the fact that Bonazzi et al. (2019) were able to successfully back-calculate the initial entrapment conditions of the synthesis by applying the Grueneisen tensor of Murri et al. (2019) together with the measured shifts of the 128cm$^{-1}$, 464cm$^{-1}$ and 206cm$^{-1}$ For this experimect, the method of Bonazzi et al. (2019) leads even to a higher precision on the estimate of the residual pressure, as can be observed comparing the scatters of the dataset synthetized at 3GPa, 775°C in Fig. 5a of Reynard and Zhong and Fig.4a of Bonazzi et al. (2019). This would rule out the presence of systematic uncertainties in the measurements performed by Bonazzi et al. (2018). Moreover, Bonazzi et al. (2019) report that the dataset Alm-2, which gives results more consistent with the calibration of Reynard and Zhong, was analyzed with the same procedure and instrument as Alm-1. Again, the impression is that the dataset Alm-1 is discarded because of the anticorrelation of the proposed sigma128 and sigma464 calibrations, which leads to incompatible values of differential stress obtained from the two calibrations. This could perhaps be caused by the complex stress states of those inclusions that cannot be modeled with the current calibration. However, I find misleading that the problem is reduced to "systematic uncertainties in the Raman peak positions" of Bonazzi et al. (2018) simply because those inclusions cannot be explained with the current calibration.

- **We do not discard the Alm-1 data, we just state that they have some internal inconsistency revealed by our analysis. A slight systematic error in the 206 peak position of 0.5 cm-1 could well explain why the Alm-2 data lie symmetrically from the 1:1 line defined by our calibration, and why the Alm-1 data are systematically offset. That kind of issue that could arise from drift of the instrument, which is unlikely given data were calibrated by Bonazzi et al. (2019). We propose this may be due to other effects such as systematic interference with garnet peak. Therefore we expanded the discussion of this particular point, and point out the sensitivity of the data analysis to such systematic deviation (lines 279-281).**

- Line 251: I find not accurate the statement that the results of the two calibrations are self-consistent. Most of the inclusions in Fig. 5c and 6b fall along a line that forms a high angle to the 1:1 line. This implies that the two calibrations give different results for the same inclusion. By just considering the dataset from Syros, for 16 inclusions out of 21 the two calibrations predict a differential stress of opposite sign. In the samples from Papua Nuova Guinea, this happens for

more than 40 inclusions out of 92. This rate is very high also for the inclusions in the synthetic datasets of Bonazzi et al. (2019). In my opinion the authors should have in mind the future users of such calibration for whom such discrepancies may be confusing, and explain the reason of the anticorrelation between the two calibrations which leads to this inconsistencies.

- **The comment was addressed in a response to a former comment on similar topics. We feel the new paragraph and supplementary figure on sensitivity analysis will help future users to assess the origin of the apparent scattering and anticorrelation.**

- Lines 254-256. I suggest that this statement is explained in more detail. If they assume that the differential stress in natural inclusions is relaxed by inelastic processes, then they should expect to measure a differential stress in natural inclusions that is less than what expected from their purely elastic model. But the entire manuscript revolves around using the measured residual pressure and the residual differential stress to estimate the entrapment conditions with an elastic model, neglecting the inelastic relaxation. And at lines 262-264 they say that the measured differential stress agrees with the predictions of elastic models, which apparently rules out any inelastic relaxation.

- **We agree with the referee. We are discussing here the measured residual stress, which are accurately described by the elastic model, and point out this observation. As some inelastic relaxation likely occurred in natural inclusions, it would require modeling for estimating the effect on the reconstructed initial P-T conditions. On second thought, the slower growth rates in natural systems than in experiments likely account for this effect. A detailed examination of such effects is clearly out of the scope of the present article. We modified the sentence (lines 300-303) to mention this possibility.**

- Lines 259-261. I believe that the statement that the high rejection rate of Gonzalez et al. (2019) is due to the discrepancies of the Grueneisen tensor is misleading. The rejection rate is determined by a different choice of criterion and threshold in the present work and in Gonzalez et al. (2019). Since the Gruneisen tensor does not have the problem of giving two different stress states for the same inclusion, the criterion in Gonzalez et al. (2019) was based on the absolute value of the differential strain of the inclusions. Gonzalez et al. (2019) applied a restrictive threshold of 1e-3 on differential strain to obtain a less dispersed distribution. However, even not applying any limiting threshold on the data of Gonzalez et al. (2019) the distributions of residual pressures of all 92 inclusions of Gonzalez et al. (2019) (see their supplementary materials) and of Reynard and Zhong (see their supplementary table, pressure obtained with the P206 calibration) are very similar, and in particular have the same standard error of 0.01 GPa. The only difference is a shift of the entire distribution of Gonzalez et al. (2019) to slightly higher pressures because of the use of different elastic properties for quartz. This shows that the high number of rejected inclusions in Gonzalez et al. (2019) is not due to problems with the Grueneisen tensor, but rather to the choice of a restrictive threshold to obtain higher accuracy on the average pressure.

- **The present calibration does not use the elastic data of quartz, since Raman peak positions are directly calibrated against pressure and stress. Only the Gruneisen formulation uses strains that need to be converted into stresses using elastic constants. The difference in absolute pressure between the two calibrations is due to the discrepancies between the stress-induced shifts obtained here experimentally and those implied in the Gruneisen analysis (Murri et al. 2019). We have added a sentence lines 231-234 to state this difference in the two approaches. For the sake of simplicity, we have suppressed the discussion on the Papua data and just state that the difference in rejection rate may arise from different criteria (line 307).**

- Table 1. As discussed above, the experiments under uniaxial stress in the *x-y* plane produce a symmetry breaking, and the two set of shift observed for the 128 cm-1 peak must not be labeled as TO-LO (Shapiro & Axe, 1972). Moreover, the note that "the TO-LO splitting was not modeled" in the calculations of Murri et al. (2019) is not correct as explained in previous comments. The LO-TO components for the 128 cm-1 peak were determined by Murri et al. (2019), but since their splitting is negligible, the Grueneisen tensor was calculated only for the TO component. Therefore, the discrepancy between the and  for the Raman modes determined in this work and those of Murri et

al. (2019) may be partly due to the symmetry breaking induced during the experiments presented here.

- **The statements regarding TO-LO splitting have been corrected.**

- Fig. 5, lines 423-424. The statement concerning the anti-correlated uncertainties is not clear. I suggest that the reason for this anticorrelation is explained in more details here or in the text (see also above comments).
- **This is now explained in the paragraph added above section 6.1**

- Fig 1: in order to represent without ambiguity the orientation adopted in the measurements, I recommend adding the corresponding Porto's notation (Gregora, 2013; Scott & Porto, 1967) defined as following:

A(BC)D

With:

A = direction of the incident radiation (wave-vector)

B = polarization of incident radiation

C = polarization of scattered radiation

D = direction of the scattered radiation (i.e. wave-vector)

where the direction and polarization of the incident and scattered radiation are defined with respect to the crystallographic axes of quartz.

**No polarizer was put in the scattered light path, thus only the polarization direction of the incident light is given, as now explained in the caption, where a sentence was added (also in section 2) to mention the use of a back-scattering geometry.**

**Technical corrections**

- Equation (12) at line 196. I suspect that the last sigma in the right side of the equation is a left over and should be removed.
- Fig 3c: the label of the y axis is missing
- 6a. I suggest to add a legend to identify the large and small symbols

**the error in eq 12 was corrected, figures and captions modified.**

**Literature cited**

Anastassakis, E. M. (1980). CHAPTER 3—Morphic Effects in Lattice Dynamics. In G. K. Horton & A. A. Maradudin (Eds.), *Dynamical Properties of Solids* (Vol. 4, pp. 157–375). North-Holland. https://doi.org/10.1016/B978-0-444-85315-8.50007-4

Angel, R. J., Murri, M., Mihailova, B., & Alvaro, M. (2019). Stress , strain and Raman shifts. *Zeitschrift Für Kristallographie-Crystalline Materials.*, *234*(2), 129–140. https://doi.org/10.1515/zkri-2018-2112

Angel, R., Mazzucchelli, M., Gonzalez-Platas, J., & Alvaro, M. (2021). A self-consistent approach to describe unit-cell-parameter and volume variations with pressure and temperature. *Journal of Applied Crystallography*, *54*(6), 1621–1630. https://doi.org/10.1107/S1600576721009092

Briggs, R. J., & Ramdas, A. K. (1977). Piezospectroscopy of the Raman spectrum of α-quartz. *Physical Review B*, *16*(8), 3815.

Gonzalez, J. P., Thomas, J. B., Baldwin, S. L., & Alvaro, M. (2019). Quartz-in-garnet and Ti-in-quartz thermobarometry: Methodology and first application to a quartzofeldspathic gneiss from eastern Papua New Guinea. *Journal of Metamorphic Geology*, *37*(9), 1193–1208. https://doi.org/10.1111/jmg.12508

Gregora, I. (2013). Raman scattering. In *International Tables for Crystallography* (pp. 334–348). John Wiley & Sons, Ltd. https://doi.org/10.1107/97809553602060000913

Murri, M., Alvaro, M., Angel, R. J., Prencipe, M., & Mihailova, B. D. (2019). The effects of non-hydrostatic stress on the structure and properties of alpha-quartz. *Physics and Chemistry of Minerals*, *46*(5), 487–499. https://doi.org/10.1007/s00269-018-01018-6

Murri, M., Gonzalez, J. P., Mazzucchelli, M. L., Prencipe, M., Mihailova, B. D., Angel, R. J., & Alvaro, M. (2022). The role of symmetry-breaking strains on quartz inclusions in anisotropic hosts: Implications for Raman elastic geobarometry. *Lithos*. https://doi.org/in press

Nye, J. F. (1985). *Physical properties of crystals: Their representation by tensors and matrices*. Oxford University Press.

Scott, J. F., & Porto, S. P. S. (1967). Longitudinal and Transverse Optical Lattice Vibrations in Quartz. *Physical Review*, *161*(3), 903–910. https://doi.org/10.1103/PhysRev.161.903

Shapiro, S. M., & Axe, J. D. (1972). Raman Scattering from Polar Phonons. *Physical Review B*, *6*(6), 2420–2427. https://doi.org/10.1103/PhysRevB.6.2420

Tekippe, V. J., Ramdas, A. K., & Rodriguez, S. (1973). Piezospectroscopic Study of the Raman Spectrum of $\ensuremath{\alpha}$-Quartz. *Physical Review B*, *8*(2), 706–717. https://doi.org/10.1103/PhysRevB.8.706

Wang, J., Mao, Z., Jiang, F., & Duffy, T. S. (2015). Elasticity of single-crystal quartz to 10 GPa. *Physics and Chemistry of Minerals*, *42*(3), 203–212. https://doi.org/10.1007/s00269-014-0711-z

---

## Author Comment (AC3)

We thank Ross Angel for his careful reading and detailed comments on our preprint. We have addressed part of his comments in our response to anonymous referee #2, in particular those concerning symmetry reduction on compression perpendicular to the *c*-axis, and we will reply here only to specific points raised by Dr Angel.

A disagreement exists around considering whether the Grüneisen parameters ($\gamma$) are constant or not. Dr Angel developed on a theory (Grüneisen) where $\gamma$ is constant, hence Raman frequencies scale to lattice parameter variation or strain through a single parameter. We have shown here and agree that this is correct for one of the quartz Raman peak (464 cm$^{-1}$), and also show that is not correct for the two other major peaks at 128 and 206 cm$^{-1}$ (Fig. 3b). Thus we fitted the relative frequency vs volume data for these two modes with a non-constant strain dependent $\gamma$. This was overlooked by Dr Angel in his re-analysis of our data, using our ambient pressure value noted as $\gamma_0$ as a constant $\gamma$ or the sum of axial g that we assumed constant. We reproduce here the section of our text where the fit was explained: "*Grüneisen plot (Fig. 3b) shows that the assumption of constant Grüneisen parameters (Angel et al., 2019) is not valid in quartz, except for the 464 cm$^{-1}$ peak. Grüneisen parameter ($\gamma$) dependence on volume is taken into account in fitting the data (Table 1) with two parameters (Reynard et al., 2012), the ambient pressure value $\gamma_0$, and its volume dependence q expressed as q=($\partial ln\gamma/\partial lnV$).*" One must include the *q* parameter to reproduce the data accurately. We have added a sentence to emphasize the difference in data analysis lines 171-172.

Constant $\gamma$ were contradicted by experiments on quartz and on other materials such as olivines (Reynard et al., 2012; Gillet et al., 1997). This does not mean that we think the Grüneisen approach through strains is less valid than the one we use here through stresses, only that it should use accurate fits with variable $\gamma$ when necessary.

Grüneisen parameters are of practical use when DFT calculations are performed at fixed strains. Here strains are inconvenient because experimental calibrations directly give pressure and stress (an experimentally determined parameter and the geologically relevant one) as a function of Raman frequencies (the other measurable variable). With the relationships we establish, we can analyze directly the effects of statistical fluctuations in the Raman measurements. We show that the minute uncertainties of 0.5 cm$^{-1}$ can have a strong effect on the determined stresses, or equivalently on the determined strains if one prefers that approach. We provide supplementary figures with sensitivity tests to show how errors propagate. Using the constant Grüneisen parameters proposed by Dr Angel and coworkers gives a fair fit to the experimental data given that only theoretical DFT data were used. However the low-pressure data essential for applications to natural inclusions are missed by up to 1.5 cm$^{-1}$, introducing systematic deviations that are larger than the uncertainties on the Raman measurements.

Thus we preferred an approach that stays close to the measured quantities in our experiments, and propose an analysis that allows detecting and evaluating systematic deviations in the Raman data rather than obtaining an averaged value of pressure and deviatoric stress for each data point. A paragraph was added lines 235-252 that describes the approach, and at the beginning of the concluding remarks (lines 324-328). Accurate average pressure and deviatoric stress are obtained by showing that populations of inclusions belong to a single population formed at similar conditions, and averaging over this population (Fig. 6, suppl Fig. 2).

Angel, R. J., Murri, M., Mihailova, B., and Alvaro, M.: Stress, strain and Raman shifts, 234, 129-140, doi:10.1515/zkri-2018-2112, 2019.

Gillet, P., Daniel, I., and Guyot, F.: Anharmonic properties of $Mg_2SiO_4$-forsterite measured from the volume dependence of the Raman spectrum, European Journal of Mineralogy, 9, 255-262, 1997.

Reynard, B., Montagnac, G., and Cardon, H.: Raman spectroscopy at high pressure and temperature for the study of the Earth's mantle and planetary materials, in: EMU Notes in Mineralogy, Vol. 12, edited by: Dubessy, J., Caumon, M., and Rull, F., 12, 365-388, 10.1180/EMU-notes.12.10, 2012.